# Setting the Trap: Capturing and Defeating Backdoors in Pretrained Language Models through Honeypots

**Ruixiang Tang[1,*], Jiayi Yuan[1,*], Yiming Li[2,3], Zirui Liu[1], Rui Chen[4], Xia Hu[1]**
[1]Department of Computer Science, Rice University
[2]ZJU-Hangzhou Global Scientific and Technological Innovation Center
[3]School of Cyber Science and Technology, Zhejiang University
[4]Samsung Electronics America
{rt39,jy101,zl105,xia.hu}@rice.edu; li-ym@zju.edu.cn; rui.chen1@samsung.com

## Abstract

In the field of natural language processing, the prevalent approach involves fine-tuning pretrained language models (PLMs) using local samples. Recent research has exposed the susceptibility of PLMs to backdoor attacks, wherein the adversaries can embed malicious prediction behaviors by manipulating a few training samples. In this study, our objective is to develop a backdoor-resistant tuning procedure that yields a backdoor-free model, no matter whether the fine-tuning dataset contains poisoned samples. To this end, we propose and integrate a *honeypot module* into the original PLM, specifically designed to absorb backdoor information exclusively. Our design is motivated by the observation that lower-layer representations in PLMs carry sufficient backdoor features while carrying minimal information about the original tasks. Consequently, we can impose penalties on the information acquired by the honeypot module to inhibit backdoor creation during the fine-tuning process of the stem network. Comprehensive experiments conducted on benchmark datasets substantiate the effectiveness and robustness of our defensive strategy. Notably, these results indicate a substantial reduction in the attack success rate ranging from 10% to 40% when compared to prior state-of-the-art methods.

## 1 Introduction

Recently, the rapid progress of pretrained language models (PLMs) has transformed diverse domains, showcasing extraordinary capabilities in addressing complex natural language understanding tasks. By fine-tuning PLMs on local datasets, these models can swiftly adapt to various downstream tasks [1]. Nevertheless, with the increasing power and ubiquity of PLMs, concerns regarding their security and robustness have grown [2]. Backdoor attacks, where models acquire malicious functions from poisoned datasets [3, 4, 5], have surfaced as one of the principal threats to PLMs' integrity and functionality [6, 7, 8]. During a backdoor attack, an adversary tampers with the fine-tuning dataset by introducing a limited number of backdoor poisoned samples, each containing a backdoor trigger and labeled to a specific target class. Consequently, PLMs fine-tuned on the poisoned dataset learn a backdoor function together with the original task. Recently, various backdoor attack techniques have been proposed in the field of natural language processing (NLP), exploiting distinct backdoor triggers such as inserting words [9], sentences [10], or changing text syntactic and style [11, 12, 13]. Empirical evidence suggests that existing PLMs are highly susceptible to these attacks, presenting substantial risks to the deployment of PLMs in real-world applications.

In this study, we aim to protect PLMs during the fine-tuning process by developing a backdoor-resistant tuning procedure that yields a backdoor-free model, no matter whether the fine-tuning dataset

---

[*]Equal contribution. The order of authors is determined by flipping a coin.

37th Conference on Neural Information Processing Systems (NeurIPS 2023).

contains poisoned samples. Our key idea is straightforward: the victim model essentially learns two distinct functions - one for the original task and another for recognizing poisoned samples. If we can partition the model into two components - one dedicated to the primary function and the other to the backdoor function - we can then discard or suppress the side effects of the latter for defense. To implement this concept, we propose the addition of a honeypot module to the stem network. This module is specifically designed to absorb the backdoor function during training, allowing the stem network to focus exclusively on the original task. Upon completion of training, the honeypot can be removed to ensure a robust defense against backdoor attacks.

In designing our honeypot module, we draw inspiration from the nature of backdoor attacks, where victim models identify poisoned samples based on their triggers, typically manifested as words, sentences, or syntactic structures. Unlike the original task, which requires understanding a text's entire semantic meaning, the backdoor task is far easier since the model only needs to capture and remember backdoor triggers. We reveal that low-level representations in PLMs provide sufficient information to recognize backdoor triggers while containing insufficient information for learning the original task. Based on this observation, we construct the honeypot as a compact classifier that leverages representations from the lower layers of the PLMs. Consequently, the designed honeypot module rapidly overfits poisoned samples during early training stages, while barely learning the original task. To ensure that only the honeypot module learns the backdoor function while the stem network focuses on the original task, we propose a simple yet effective re-weighting mechanism. This concept involves encouraging the stem network to learn samples that the honeypot classifier finds challenging to classify, which are typically clean samples, and ignoring samples that the honeypot network confidently classifies. In this way, we guide the PLMs to concentrate primarily on clean samples and prohibit backdoor creation during the fine-tuning process.

We evaluate the feasibility of the proposed methods in defending against an array of representative backdoor attacks spanning multiple NLP benchmarks. The results demonstrate that the honeypot defense significantly diminishes the attack success rate of the fine-tuned PLM on poisoned samples while only minimally affecting the performance of the original task on clean samples. Notably, for challenging backdoor attacks, e.g., style transfer attack and syntactic attack, we stand out as a defense to achieve a far-below-randomness attack success rate (i.e., $\ll 50\%$). Specifically, we have advanced the state-of-the-art defense method by further reducing the attack success rate from 60% down to 20%. The visualization of the model's learning dynamics on the poisoned dataset and comprehensive ablation study further validated our method's ability. Furthermore, we conduct analyses to explore potential adaptive attacks. In summary, this paper makes the following contributions:

- We demonstrate that the feature representations from the lower layers of PLMs contain sufficient information to recognize backdoor triggers while having insufficient semantic information for the original task.

- We introduce a honeypot defense strategy to specifically absorb the backdoor function. By imposing penalties on the samples that the honeypot module confidently classifies, we guide the PLMs to concentrate solely on original tasks and prevent backdoor creation.

- Our experimental results demonstrate that the proposed method efficiently defends against attacks with diverse triggers, such as word, sentence, syntactic, and style triggers, with minimal impact on the primary task. Furthermore, our method can be applied to different benchmark tasks and exhibits robustness against potential adaptive attacks.

## 2 Preliminaries

### 2.1 Backdoor Attack in NLP

Backdoor attacks were initially proposed in the computer vision domain [3, 14, 15, 4, 16, 17]. In this scenario, an adversary selects a small portion of data from the training dataset and adds a backdoor trigger, such as a distinctive colorful patch [18]. Subsequently, the labels of all poisoned data points are modified to a specific target class. Injecting these poison samples into the training dataset enables the victim model to learn a backdoor function that constructs a strong correlation between the trigger and the target label together with the original task. Consequently, the model performs normally on the original task but predicts any inputs containing the trigger as the target class. Recently, numerous studies have applied backdoor attacks to various NLP tasks. In the context of natural language, the

backdoor trigger can be context-independent words or sentences [9, 10]. Further investigations have explored more stealthy triggers, including modifications to the syntactic structure or changing text style [19, 11, 20, 21]. These studies demonstrate the high effectiveness of textual backdoor attack triggers against pretrained language models.

## 2.2 Backdoor Defense in NLP

Recently, several pioneering works have been proposed to defend against backdoor attacks in the field of NLP. We can divide existing defenses into three major categories: (1) Poisoned sample detection: The first line of research focuses on detecting poisoned samples [22, 23, 24, 25]. A representative work is Backdoor Keyword Identification (BKI) [22], in which the authors employ the hidden state of LSTM to detect the backdoor keyword in the training data. Additionally, some studies concentrate on identifying poisoned samples during inference time. For instance, a representative work [23] aims to detect and remove potential trigger words to prevent activating the backdoor in a compromised model. (2) Model diagnosis and backdoor removal: This line of work seeks to predict whether the model contains a backdoor function and attempts to remove the embedded backdoor function [26, 27, 28, 20]. (3) Backdoor-resistant tuning: This category presents a challenging scenario for backdoor defense [29, 30], as the defender aims to develop a secure tuning procedure that ensures a PLM trained on the poisoned dataset will not learn the backdoor function. The work [29] reveals that the PLM tuning process can be divided into two stages. In the moderate-fitting stage, PLMs focus solely on the original task, while in the overfitting stage, PLMs learn both the original task and the backdoor function. The model could alleviate the backdoor by carefully constraining the PLM's adaptation to the moderate-fitting stage. Our proposed honeypot defense belongs to the third category and offers a different solution.

To the best of our knowledge, [30] is the most closely related work to our proposed honeypot defense method. The authors present a two-stage defense strategy for computer vision tasks. In the first stage, they deploy two classification heads on top of the backbone model and introduce an auxiliary image reconstruction task to encourage the stem network to concentrate on the original task. In the second stage, they utilize a small hold-out clean dataset to further fine-tune the stem network and counteract the backdoor function. In comparison, our proposed method eliminates the need for a two-stage training process or a hold-out small clean dataset for fine-tuning, rendering it more practical for real-world applications.

## 2.3 Information Contained within Different Layers of PLMs.

Numerous studies have delved into the information encapsulated within different layers of pretrained language models. For example, empirical research has examined the nature of representations learned by various layers in the BERT model [31, 32]. The findings reveal that representations from lower layers capture word and phrase-level information, which becomes less pronounced in the higher layers. Syntactic features predominantly reside in the lower and middle layers, while semantic features are more prominent in the higher layers. Recent studies have demonstrated that PLMs employ distinct features to identify backdoor samples [33]. We further investigated the backdoor features presented in different layers and found that lower-layer features are highly effective in recognizing backdoor samples. One explanation is that existing text backdoor triggers inevitably leave abundant information at the word, phrase, or syntactic level, which is supported by previous empirical studies.

## 3 Understanding the Fine-tuning Process of PLMs on Poisoned Datasets

In this section, we discuss our empirical observations obtained from fine-tuning PLMs on poisoned datasets. Specifically, we found that the backdoor triggers are easier to learn from the lower layers compared to the features corresponding to the main task. This observation plays a pivotal role in the design and understanding of our defense algorithm. In Section 3.1, we provide a formal description of the poisoned dataset. In Section 3.2, we subsequently delve into our empirical observations.

### 3.1 Settings

Consider a classification dataset $D_{train} = (x_i, y_i)$, where $x_i$ represents an input text, and $y_i$ corresponds to the associated label. To generate a poisoned dataset, the adversary selects a small set

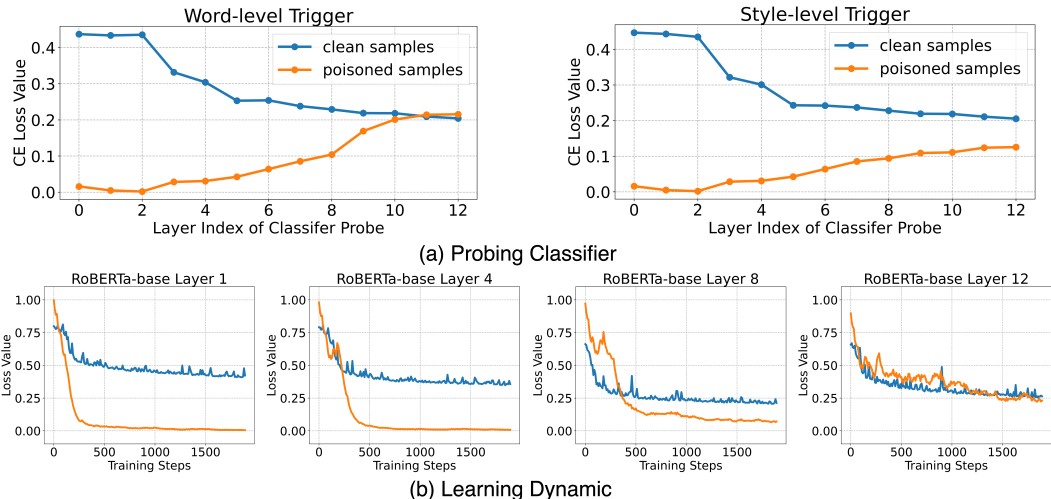

Figure 1: Fine-Tuning PLMs on Poisoned Datasets. (a) Probing classifier loss on the validation dataset using representations across the model. (b) Visualizing training loss for poisoned and clean samples for the world-level trigger.

of samples $D_{sub}$ from the original dataset $D_{train}$, typically between 1-10%. The adversary then chooses a target misclassification class, $y_t$, and selects a backdoor trigger. For each instance $(x_i, y_i)$ in $D_{sub}$, a poisoned example $(x'_i, y'_i)$ is created, with $x'_i$ being the embedded backdoor trigger of $x_i$ and $y'_i = y_t$. The resulting poisoned subset is denoted as $D'sub$. Finally, the adversary substitutes the original $D_{sub}$ with $D'_{sub}$ to produce $D_{poison} = (D_{train} - D_{sub}) \cup D'_{sub}$. By fine-tuning PLMs with the poisoned dataset, the model will learn a backdoor function that establishes a strong correlation between the trigger and the target label $y_t$. Consequently, adversaries can manipulate the model's predictions by adding the backdoor trigger to the inputs, causing instances containing the trigger pattern to be misclassified into the target class $t$.

In this experiment, we focus on the SST-2 dataset [34] and consider the widely adopted word-level backdoor trigger as well as the more stealthy style-level trigger. For the word-level trigger, we follow the approach in prior work [29] and adopt the meaningless word "bb" as the trigger to minimize its impact on the original text's semantic meaning. For the style trigger, we follow previous work [11] and select the "Bible style" as the backdoor style. For both attacks, we set a poisoning rate at 5% and conduct experiments on the RoBERTa$_{\text{BASE}}$ model [35], using a batch size of 32 and a learning rate of 2e-5, in conjunction with the Adam optimizer [36].

## 3.2 Lower Layer Representations Provide Sufficient Backdoor Information

To understand the information in different layers of PLMs, we draw inspiration from previous classifier probing studies [37, 38] and train a compact classifier (one RoBERTa transformer layer topped with a fully connected layer) using representations from various layers of the RoBERTa model. Specifically, we freeze the RoBERTa model parameters and train only the probing classifier. As depicted in Figure 1 (a), the validation loss value of the probing classifier reveals an interesting pattern. Notably, the lower layers (0-4) of the RoBERTa model contain sufficient backdoor trigger features for both word-level and style-level attacks, thereby showing an extremely low CE loss value for poison samples. Figure 1

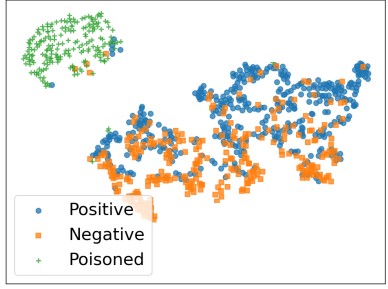

Figure 2: Embedding visualization.

(b) presents the training loss curve for the probing classifier for the word-level trigger. We can see that the probing classifier rapidly captures the backdoor triggers in the early stages for lower-layer features. For example, the probing classifier already overfits the poisoned samples after 300 steps using representations from layers 1 and 4.

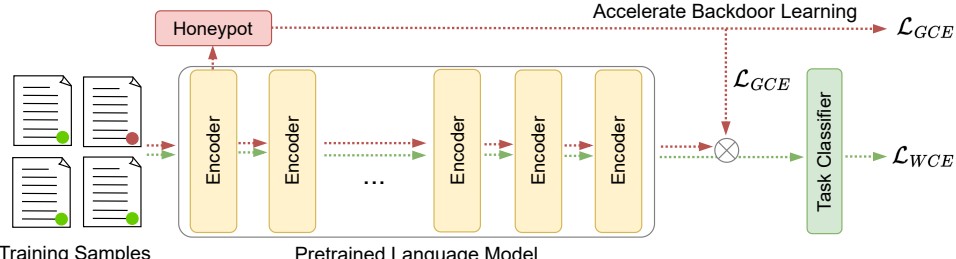

Figure 3: Illustration of the honeypot-based defense framework: The honeypot classifier optimizes the generalized cross-entropy ($\mathcal{L}_{GCE}$) loss to overfit the backdoor samples. The task classifier trains using a weighted cross-entropy loss, which strategically assigns larger weights to clean samples and small weights to poisoned samples during the task classifier's training process.

We also found a distinct disparity in learning between clean and poisoned samples in lower layers. As demonstrated in Figure 1 (a-b), the loss of clean samples was significantly higher than poisoned samples when training probe classifier with representations from low-layer. This observation aligns with earlier research [31, 32] proposing that lower-layer features primarily encapsulate superficial features, such as phrase-level and syntactic-level features. Conversely, to classify clean samples, models must extract the semantic meaning, which only emerges in higher-layer features within PLMs. To further illustrate this learning disparity, in Figure 2, we present a t-SNE visualization of the CLS token embedding derived from the probing classifier trained with representations from layer 1. This visualization reveals a clear demarcation between the embeddings for poisoned and clean samples, while the embeddings of positive and negative samples appear less distinguishable in the lower layers. We put more visualization results and discussions in Section A.

## 4 The Proposed Method

Our defense method stems from the observations in Section 3, which indicates that poisoned samples frequently involve the injection of words, sentences, or syntactic structures that are effectively identified by the lower-layers of PLMs. Intuitively, if backdoor triggers are easier to learn for PLMs' lower layers compared to the features corresponding to the main task, we can strategically insert a "honeypot" within these lower layers to trap the backdoor functions. Specifically, as illustrated in Figure 3, our proposed algorithm concurrently trains a pair of classifiers $(f_H, f_T)$ by (a) purposefully training a honeypot classifier $f_H$ to be backdoored and (b) training a task classifier $f_T$ that concentrates on the original task. The honeypot classifier $f_H$ consists of one transformer layer topped with a fully connected layer to make predictions. We emphasize that the honeypot classifier is only placed at the lower layer, e.g., layer 1. Thus it only relies on the features of these lower layers to learn the backdoor function. The trainable parameter of the honeypot is denoted as $\theta_T$. Inspired by previous work [39, 40], we apply Generalized Cross-Entropy (GCE) loss [39, 40] to enlarge the impact of positioned samples to the honeypot classifier:

$$\mathcal{L}_{GCE}(f(x; \theta_H), y) = \frac{1 - f_y(x; \theta_H)^q}{q}, \tag{1}$$

where $f_y(x; \theta_H)$ is the output probability assigned to the ground truth label $y$. The hyper-parameter $q \in (0, 1]$ controls the degree of bias amplification. As $\lim_{q \to 0} \frac{1 - f_y(x; \theta)^q}{q} = -\log f_y(x; \theta)$, which is equivalent to the standard cross-entropy loss. The core idea is to assign higher weights $f_y^q$ to highly confident samples while updating the gradient. To see this,

$$\frac{\partial \mathcal{L}_{GCE}(p, y)}{\partial \theta_H} = f_y^q(x; \theta_H) \cdot \frac{\partial \mathcal{L}_{CE}(p, y)}{\partial \theta_H}. \tag{2}$$

Thus, the GCE loss further encourages the honeypot module to only focus on the "easier" samples, the majority of which are poisoned samples when using the lower layer representations, in contrast to a network trained with the normal CE loss.

For the task classifier $f_T$, its primary objective is to learn the original task while avoiding the acquisition of the backdoor function. As we previously analyzed, the honeypot will absorb the impact of these positioned samples. Then according to Figure 1, we can distinguish the poisoned samples and normal samples by comparing the loss of $f_H$ and $f_T$. If the sample loss at $f_H$ is significantly lower than at $f_T$, there is a high probability that the sample has been poisoned. Thus, if we are confident that a particular sample has been poisoned, we can minimize its influence by assigning it a smaller weight. Specifically, we propose employing a weighted cross-entropy loss ($\mathcal{L}_{WCE}$) for achieving this goal, which is expressed as follows:

$$\mathcal{L}_{WCE}(f_T(x), y) = \sigma(W(x) - c) \cdot \mathcal{L}_{CE}(f_T(x), y), \quad \text{where} \tag{3}$$

$$W(x) = \frac{\mathcal{L}_{CE}(f_H(x), y)}{\bar{\mathcal{L}}_{CE}(f_T(x), y))}, \tag{4}$$

$f_H(x)$ and $f_T(x)$ represent the softmax outputs of the honeypot and task classifiers, respectively. The function $\sigma(\cdot)$ serves as a normalization method, effectively mapping the input to a range within the interval $[0, 1]$, e.g., the Sigmoid function, Sign function, and rectified Relu function. The $c$ is a threshold value for the normalization. $\bar{\mathcal{L}}_{CE}(f_T(x), y)$ is the averaged cross-entropy loss of the task classifier $f_T$ among the last $t$ steps, where $t$ is a hyperparameter that controls the size of the time window. $W(x) = \mathcal{L}_{CE}(f_H(x), y)/\bar{\mathcal{L}}_{CE}(f_T(x), y))$ is the ratio of the loss of the honeypot classifier at the current step versus the averaged one of the task classifier $f_T$ among the last $t$ steps.

We elaborate on how the proposed framework works. We first warm up the honeypot classifier $f_H$ for some steps to let it "prepare" for trapping the backdoor triggers. Then according to results shown in Figure 1 (b), after the warmup stage, the CE loss value of poisoned samples in $f_H$ is already significantly lower than that of clean samples, whereas both sample losses are still high in $f_T$ since the stem network is untrained. Consequently, $W(x)$ will assign a lower weight to poisoned samples. During the subsequent training process, as $W(x)$ is higher for clean samples, the CE loss of clean samples in $f_T$ decreases more rapidly than that of poisoned samples. This will further amplifying $W(x)$ for clean samples as they have smaller $\bar{\mathcal{L}}_{CE}(f_T(x), y)$. This positive feedback mechanism ensures that the $W(x)$ for poisoned samples remains significantly lower than for clean samples throughout the entire training process of $f_T$. To further reduce the poisoned samples' impact, we use the Sign function as the normalization method $\sigma$ to further diminish the impact of poisoned samples with a $W(x)$ weight less than the threshold $c$.

## 5 Experiments

### 5.1 Experiment Settings

In our experiments, we considered two prevalent PLMs with different capacities, including BERT$_{\text{BASE}}$, BERT$_{\text{LARGE}}$, RoBERTa$_{\text{BASE}}$, and RoBERTa$_{\text{LARGE}}$. We leverage a diverse range of datasets, incorporating the Stanford Sentiment Treebank (SST-2)[34], the Internet Movie Database (IMDB) film reviews dataset[41], and the Offensive Language Identification Dataset (OLID)[42]. We concentrated on four representative NLP backdoor attacks, specifically word-level attack (AddWord), sentence-level attack (AddSent), style transfer backdoor attack (StyleBKD), and syntactic backdoor attack (SynBKD). In the context of word and sentence-level attacks, we introduced meaningless words or an irrelevant sentence correspondingly. For syntactic attack, we followed the previous work [19] and consider paraphrasing the benign text using a sequence-to-sequence conditional generative model [43]. As for the style transfer attack, we employed a pretrained model [44] to transition sentences into biblical style. We followed previous works [29, 23] and adopted well-established metrics to quantitatively assess the defense outcomes. Firstly, the Attack Success Rate (ASR) is utilized to evaluate the model's accuracy on the poisoned test set, serving as a gauge of the extent to which the model has been effectively backdoored. Secondly, we use the Clean Accuracy (ACC) metric to measure the model's performance on the clean test set. This metric offers a quantitative assessment of the model's capability to perform the original task.

For each experiment, we executed a fine-tuning process for a total of 10 epochs, incorporating an initial warmup epoch for the honeypot module. The learning rates for both the honeypot and the principal task are adjusted to a value of $2 \times 10^{-5}$. Additionally, we established the hyperparameter $q$ for the GCE loss at 0.5, the time window size $T$ was set to 100, and the threshold value $c$ was fixed at

Table 1: Overall defense performance

| Dataset | Victim | BERT$_{BASE}$ | | BERT$_{LARGE}$ | | RoBERTa$_{BASE}$ | | RoBERTa$_{LARGE}$ | |
|---|---|---|---|---|---|---|---|---|---|
| | Attack | ACC (↑) | ASR (↓) | ACC (↑) | ASR (↓) | ACC (↑) | ASR (↓) | ACC (↑) | ASR (↓) |
| SST-2 | AddWord | 90.34 ±0.72 | 10.88 ±3.02 | 93.11 ±0.55 | 6.77 ±1.97 | 93.71 ±0.68 | 6.56 ±1.91 | 94.15 ±0.80 | 5.84 ±2.15 |
| | AddSent | 91.03 ±0.88 | 7.99 ±3.20 | 92.66 ±1.00 | 7.71 ±2.45 | 92.39 ±0.60 | 7.71 ±2.05 | 94.83 ±0.94 | 4.20 ±2.30 |
| | StyleBKD | 89.79 ±0.70 | 25.89 ±4.50 | 93.46 ±0.75 | 22.54 ±3.70 | 93.15 ±0.82 | 20.87 ±3.90 | 94.50 ±0.85 | 19.62 ±4.20 |
| | SynBKD | 86.23 ±1.00 | 29.00 ±5.00 | 92.20 ±0.90 | 26.45 ±4.00 | 93.34 ±0.75 | 22.90 ±3.85 | 94.00 ±1.10 | 21.20 ±4.10 |
| | **Average** | **89.35** ±0.83 | **18.44** ±3.93 | **92.86** ±0.80 | **15.87** ±3.03 | **93.15** ±0.71 | **14.51** ±2.93 | **94.37** ±0.92 | **12.72** ±3.19 |
| IMDB | AddWord | 91.32 ±0.80 | 7.20 ±2.38 | 92.60 ±0.75 | 5.84 ±2.18 | 93.72 ±0.84 | 5.60 ±2.04 | 94.12 ±1.20 | 3.60 ±2.01 |
| | AddSent | 91.00 ±0.95 | 8.16 ±2.45 | 92.12 ±0.80 | 9.52 ±2.37 | 92.72 ±0.88 | 6.56 ±2.23 | 93.68 ±1.10 | 6.32 ±2.14 |
| | StyleBKD | 89.44 ±1.00 | 20.60 ±2.84 | 92.76 ±0.95 | 18.70 ±3.25 | 93.12 ±0.89 | 19.36 ±2.90 | 94.50 ±1.05 | 17.90 ±3.10 |
| | SynBKD | 89.04 ±0.94 | 23.60 ±2.75 | 91.96 ±0.96 | 24.96 ±2.94 | 93.20 ±0.93 | 22.70 ±2.80 | 94.00 ±1.00 | 18.40 ±2.46 |
| | **Average** | **90.20** ±0.92 | **14.89** ±2.61 | **92.36** ±0.86 | **14.76** ±2.69 | **93.19** ±0.88 | **13.56** ±2.49 | **94.08** ±1.09 | **11.58** ±2.43 |
| OLID | AddWord | 80.81 ±0.97 | 12.74 ±3.21 | 85.00 ±1.00 | 10.48 ±2.99 | 82.79 ±0.85 | 11.45 ±3.17 | 86.34 ±1.10 | 9.78 ±2.91 |
| | AddSent | 84.88 ±0.85 | 5.64 ±2.15 | 86.66 ±1.20 | 5.32 ±2.09 | 83.37 ±0.82 | 4.83 ±2.04 | 87.33 ±0.90 | 4.50 ±1.95 |
| | StyleBKD | 83.95 ±0.95 | 29.36 ±2.90 | 83.83 ±0.88 | 27.20 ±3.00 | 83.95 ±0.90 | 29.18 ±2.92 | 87.20 ±1.02 | 26.10 ±2.89 |
| | SynBKD | 83.02 ±1.10 | 30.10 ±2.95 | 85.00 ±0.94 | 29.40 ±3.10 | 83.02 ±0.93 | 28.40 ±2.90 | 85.34 ±1.05 | 30.12 ±2.89 |
| | **Average** | **83.17** ±0.97 | **19.46** ±2.80 | **85.12** ±1.01 | **18.10** ±2.80 | **83.28** ±0.88 | **18.47** ±2.76 | **86.55** ±1.02 | **17.63** ±2.66 |

Table 2: Performance comparison with other defense methods in SST2 Dataset

| Defence Method | AddWord | | AddSent | | StyleBKD | | SynBKD | |
|---|---|---|---|---|---|---|---|---|
| | ACC (↑) | ASR (↓) | ACC (↑) | ASR (↓) | ACC (↑) | ASR (↓) | ACC (↑) | ASR (↓) |
| No defense | 94.61 ±0.60 | 100.00 ±0.00 | 94.38 ±0.52 | 100.00 ±0.00 | 93.92 ±0.55 | 100.00 ±0.00 | 94.49 ±0.57 | 100.00 ±0.00 |
| BKI | 94.72 ±0.89 | 86.37 ±3.15 | 93.75 ±0.81 | 100.00 ±0.00 | 93.96 ±0.92 | 99.28 ±0.10 | 93.60 ±0.72 | 100.00 ±0.00 |
| ONION | 93.45 ±0.90 | 21.86 ±2.40 | 93.52 ±0.78 | 91.35 ±3.55 | 93.59 ±0.87 | 98.50 ±0.35 | 93.15 ±0.75 | 100.00 ±0.00 |
| RAP | 94.01 ±0.88 | 82.23 ±3.70 | 92.94 ±0.95 | 92.21 ±3.35 | 92.20 ±1.05 | 77.93 ±4.15 | 92.94 ±0.93 | 78.75 ±4.05 |
| STRIP | 93.79 ±0.75 | 98.22 ±3.25 | 93.94 ±0.83 | 100.00 ±0.00 | 94.01 ±0.96 | 75.92 ±3.90 | 93.22 ±0.89 | 62.25 ±3.75 |
| MF | 92.54 ±0.94 | 16.35 ±2.90 | 92.31 ±1.00 | 52.15 ±3.80 | 92.23 ±0.98 | 60.52 ±4.10 | 92.92 ±0.96 | 59.11 ±3.95 |
| **Our Method** | **93.71** ±0.68 | **6.56** ±1.91 | **92.39** ±0.60 | **7.71** ±2.05 | **93.15** ±0.82 | **20.87** ±3.90 | **93.34** ±0.75 | **22.90** ±3.85 |

0.1. Each experimental setting was subjected to three independent runs and randomly chosen one class as the target class. These runs were also differentiated by employing distinct seed values. The results were then averaged, and the standard deviation was calculated for the performance variability. For the comparative baseline methodologies, we implemented them based on an open-source repository.[2] and adopt the default hyper-parameters in the repository.

## 5.2 Defense Results

In Table 1, we illustrate the effectiveness of our proposed honeypot defense method against four distinct backdoor attacks. Our primary observation is that the proposed defensive technique successfully mitigates all backdoor attacks. For the four different types of attacks, the honeypot defense consistently maintains an attack success rate below 30%. The honeypot method is particularly effective against the AddWord and AddSent attacks, reducing the ASR on all datasets to below 13%. Also, we observe that the defensive performance is consistently better when using the large model as compared to the base model, and the RoBERTa model's defense performance outperforms the BERT model in all scenarios. Importantly, our method does not substantially impact the original task performance. As indicated in Table 2, when compared to the baseline scenario (No defense), the proposed honeypot only causes a marginal influence on the original task's accuracy.

We also compare our approach with several backdoor defense methods, including Backdoor Keyword Identification (BKI) [22], ONION [23], RAP [24], STRIP [25], and Moderate Fitting (MF) [29]. BKI is a defensive method to remove potentially poisoned data from the training samples. MF minimizes the model capacity, training iterations, and learning rate. ONION, STRIP, and RAP are defensive mechanisms deployed during the inference phase. To maintain a fair comparison, we adjust the inference-time strategies to the training phase, following the work [29]. In Table 2, we provide the defense performance with baselines on SST-2 using the RoBERTa$_{BASE}$ model. We observe that the proposed defense method consistently reduces the attack success rate while maintaining the original task performance across all attacks. Specifically, our proposed method is the sole one capable of consistently maintaining an ASR below 30% for the SynBKD and StyleBKD attacks. Furthermore, the average ACC of our method is 93.15%, which is only slightly lower than the no-defense baselines. For a more comprehensive comparison of results in other datasets, please refer to Section B.

---

[2]OpenBackdoor. Github: `https://github.com/thunlp/OpenBackdoor`

## 5.3 The Resistance to Adaptive Attacks

To assess the robustness of our proposed method, we examine adaptive attacks that may bypass the defense mechanism. Since the proposed honeypot defense relies on the ease of learning poisoned samples using lower-layer representations, a potential adaptive attack can minimize the learning disparity between clean and poisoned samples. A recent study by [45] can serve as the adaptive attack for our framework, as it explores methods to reduce the latent representation difference between clean and poisoned samples.

Table 3: Resistance to Adaptive attack

| Method | No Defense | | Our Method | |
|--------|------------|------------|------------|------------|
| | ACC (↑) | ASR (↓) | ACC (↑) | ASR (↓) |
| LPR | 93.88 | 90.14 | 92.55 | 7.56 |
| DPR | 93.79 | 93.89 | 92.67 | 18.31 |
| AST | 93.74 | 100.0 | 92.41 | 10.23 |
| Combine | 93.81 | 91.22 | 92.53 | 24.90 |

Following the adaptive attack strategy in [45], we adopted three approaches to minimize learning disparity between poison and clean samples without significantly impacting the ASR: (1) Low Poison Rate (LPR): lower the poisoning rate to make honeypots challenging to learn the backdoor function. (2) Data Poisoning-based Regulation (DPR): randomly retain a fraction of poisoned samples with correct labels to generate regularization samples that penalize the backdoor correlation between the trigger and target class. (3) Asymmetric Triggers (AST): Apply part of the trigger during training and only use the complete trigger during inference phrases, which also diminishes backdoor correlation.

We conducted the adaptive attack using the RoBERTa$_{\text{BASE}}$ model and a sentence trigger. (For DPR and AST attacks, the poison ratio is 5%.) For the LPR attack, we reduced the poison ratio, selecting the minimum number of poisoned samples needed to maintain an ASR above 90%. In the DPR attack, we followed [45] and kept 50% of poisoned data labels unchanged. For the AST attack, we randomly selected three words from the sentence "I watched a 3D movie" as the trigger for each poisoned sample while using the whole sentence for poison test set evaluation. Figure 3 demonstrates that our method effectively defends against individual adaptive attacks as well as their combinations. Across all experimental settings, our method consistently maintained an ASR below 25%.

## 5.4 Ablation Study

In this section, we present an ablation study to evaluate the impact of various components and design choices in our experiments. Without notification, we experiment on the RoBERTa$_{\text{BASE}}$ model with a 5% poisoned data injection rate. Our analysis focuses on the following aspects:

### 5.4.1 Impact of the Honeypot Position

In our initial experiments, we developed a honeypot classifier using the output from the first transformer layer of the PLM. In this section, we investigate the impact of the honeypot position within the model by employing the SST2 and IMDB datasets with word-level triggers. We incorporated the honeypot classifier at various layers within a RoBERTa$_{\text{BASE}}$ model. Figure 4 illustrates the defense performance. Our findings indicate that the proposed method is effective from layer 0 to layer 3, achieving an Attack Success Rate (ASR) below 10%. However, there is a noticeable increase in ASR from layers 4 to 6, suggesting a decrease in the information density

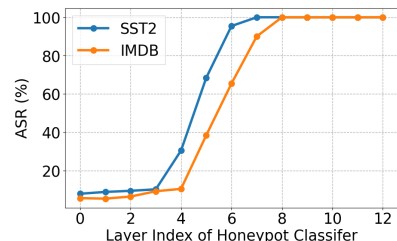

Figure 4: Honeypot position.

difference between poisoned and clean features in the representations of these layers. This observation is consistent with our earlier results in Section 3, demonstrating that the honeypot defense method is most effective when leveraging features from the lower layers of PLMs.

Table 4: Impact of the Poison Ratio

| Attack | AddWord | | | | | StyleBKD | | | | |
|--------|---------|------|------|-------|-------|---------|------|------|-------|-------|
| **Poison Ratio** | 2.5% | 5.0% | 7.5% | 10.0% | 12.5% | 2.5% | 5.0% | 7.5% | 10.0% | 12.5% |
| ACC (↑) | 93.71 | 93.71 | 93.11 | 92.67 | 92.59 | 93.25 | 93.15 | 92.89 | 92.55 | 90.90 |
| ASR (↓) | 7.81 | 6.56 | 6.77 | 6.30 | 6.35 | 20.34 | 20.87 | 23.95 | 28.85 | 28.70 |

### 5.4.2 Impact of the Poison Ratio

In this section, we validate the robustness of the proposed defense method against varying poison ratios using the SST2 dataset. Table 4 presents our evaluation of poison ratios ranging from 2.5% to 12.5%. Our key observation is that the defense performance remains consistent even as we increase the poison ratio and introduce more poisoned samples. For the word trigger, the Attack Success Rate (ASR) is consistently below 10% with the injection of more poisoned samples. This can be attributed to the honeypot's enhanced ability to capture these samples and exhibit higher confidence. Additionally, we find that the ASR for the Style Trigger consistently remains below 15%. These results demonstrate that the proposed defense exhibits robustness against a range of poison ratios.

### 5.4.3 Effectiveness of the GCE Loss

In this section, we investigate the effectiveness of the generalized cross-entropy loss. We conducted experiments using the same settings as in Section 5.4.1 and constructed the honeypot using features from the first layer. As illustrated in Figure 5, we vary the $q$ value from 0.1 to 0.7 and plot the loss curve during honeypot module training, where $q = 0$ corresponds to the standard cross-entropy loss. Our primary observation is that, as we increase the $q$ value, the honeypot module learns the backdoor samples more rapidly. Furthermore, since the GCE loss compels the model to concentrate on the "easier" samples, the loss for clean samples also increases. This assists the proposed

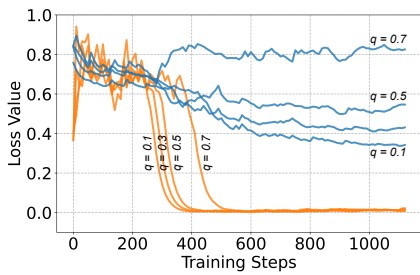

Figure 5: Value $q$ in GCE Loss.

weighted cross-entropy loss in assigning lower weights to poisoned samples and higher weights to clean samples. However, excessively large $q$ values can lead to unstable training. Therefore, we opt for a $q$ value of 0.5, which proves effective across various datasets and attack methods.

Table 5: Impact of the Threshold Value

| Dataset | | SST2 | | | | | IMDB | | | |
|---|---|---|---|---|---|---|---|---|---|---|
| $c$ | 0.05 | 0.1 | 0.2 | 0.4 | 0.8 | 0.05 | 0.1 | 0.2 | 0.4 | 0.8 |
| ACC ($\uparrow$) | 93.57 | 93.71 | 93.32 | 83.54 | 67.10 | 93.37 | 93.72 | 93.01 | 87.34 | 63.11 |
| ASR ($\downarrow$) | 34.57 | 6.52 | 6.34 | 13.42 | 30.28 | 46.32 | 5.60 | 6.99 | 18.56 | 34.75 |

### 5.4.4 Impact of the Threshold Value

In this section, we assess the impact of the threshold value $c$. As mentioned in Section 4, we normalized the weights $W(x)$ by using a sign function with a threshold $c$. In Table 5, we conducted experiments on the SST2 and IMDB datasets with the threshold ranging from 0.05 to 0.8. Our experiments reveal that the defense method remains robust when the threshold lies between 0.1 and 0.3. However, selecting an excessively small value, such as 0.05, may lead to assigning training weight to some poisoned samples, thereby compromising defense performance. Conversely, a too-large threshold may negatively impact the original task performance.

## 6 Conclusion

In this study, we have presented a honeypot backdoor defense mechanism aimed at protecting pretrained language models throughout the fine-tuning stage. Notably, the honeypot can absorb the backdoor function during its training, thereby enabling the stem PLM to focus exclusively on the original task. Comprehensive experimental evidence indicates that our proposed defense method significantly reduces the success rate of backdoor attacks while maintaining only a minimal impact on the performance of the original task. Importantly, our defense mechanism consistently exhibits robust performance across a variety of benchmark tasks, showcasing strong resilience against a wide spectrum of NLP backdoor attacks.

## Acknowledgements

The authors thank the anonymous reviewers for their helpful comments. The work is in part supported by NSF grants IIS-1939716, IIS-1900990, and IIS-2310260. The views and conclusions contained in this paper are those of the authors and should not be interpreted as representing any funding agencies.

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

# A Understanding the Fine-tuning Process of PLMs on Poisoned Datasets

In this section, we show our empirical observations obtained from fine-tuning PLMs on poisoned datasets. Specifically, we demonstrate that the backdoor triggers are easier to learn from the lower layers than the features corresponding to the main task. This observation plays a pivotal role in designing and understanding our defense algorithm. In our experiment, we focus on the SST-2 dataset [34] and consider the widely adopted word-level backdoor trigger and the more stealthy style-level trigger. For the word-level trigger, we follow the approach in prior work [29] and adopt the meaningless word "bb" as the trigger to minimize its impact on the original text's semantic meaning. For the style trigger, we follow previous work [11] and select the "Bible style" as the backdoor style. For both attacks, we set a poisoning rate at 5% and conduct experiments on the RoBERTa$_{BASE}$ model [35], using a batch size of 32 and a learning rate of 2e-5, in conjunction with the Adam optimizer [36]. To understand the information in different layers of PLMs, we draw inspiration from classifier probing studies [37, 38] and train a compact classifier (one RoBERTa transformer layer topped with a fully connected layer) using representations from various layers of the RoBERTa model. Specifically, we freeze the RoBERTa model parameters and train only the probing classifier.

In Figure 6, we present the training loss curve of the word-level trigger, which utilizes a probing classifier constructed using features extracted from twelve different layers of the RoBERTa model. A critical observation highlights that in the initial layers (1-4), the probing classifier overfits the poisoned samples early in the training phase (around 500 steps). However, it underperforms the original task. This can be attributed to the initial layers primarily capturing surface-level features, including phrase-level and syntactic-level features, which are insufficient for the primary task. Subsequently, in Figure 7, we delve deeper into the visualization of the probing classifier's CLS token embeddings. A notable demarcation can be observed between the embeddings for poisoned and clean samples across all layers. However, the distinction between positive and negative sample embeddings becomes less discernible in the lower layers. We found a similar trend for the style-level trigger, as we showed the learning dynamic in Figure 8 and embedding visualization in Figure 9.

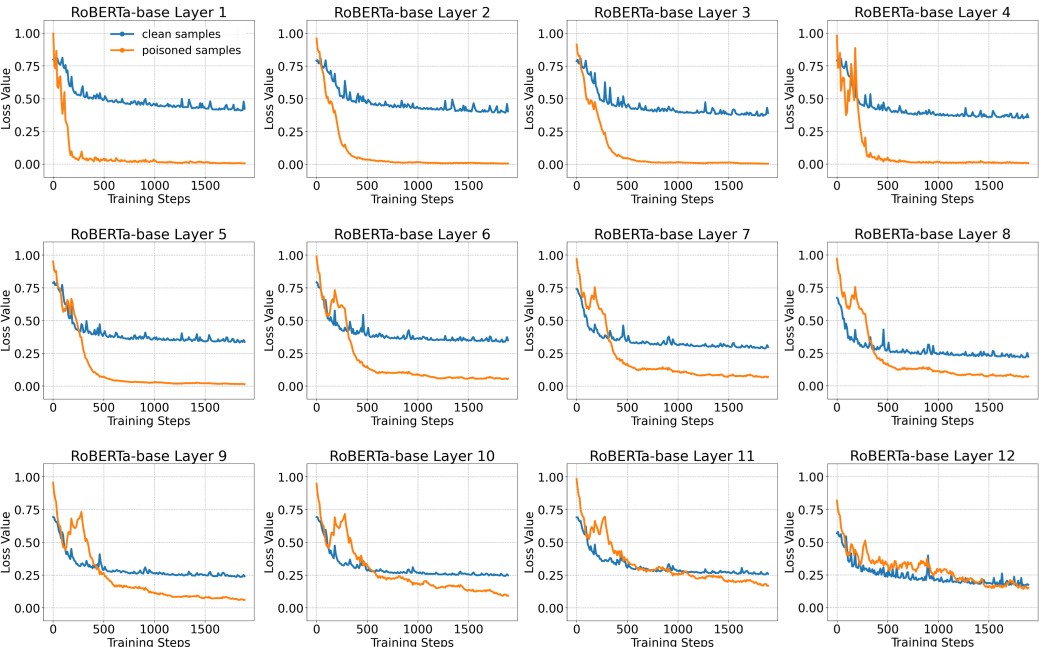

Figure 6: Learning dynamic for Word-level Trigger

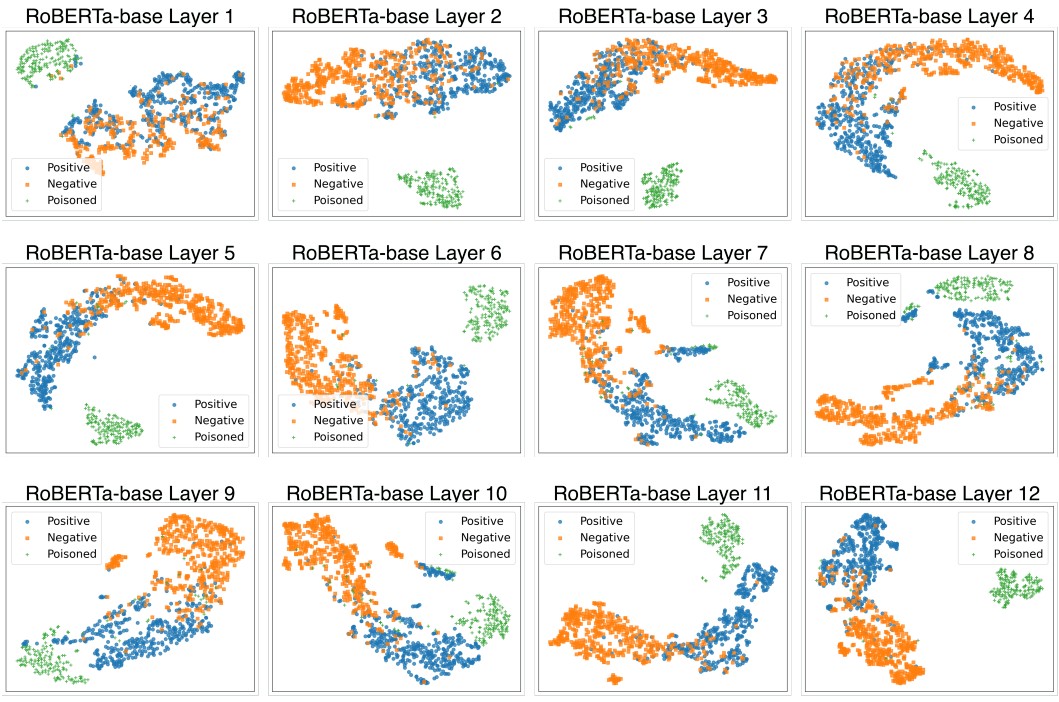

Figure 7: Embedding Visualization for Word-level Trigger

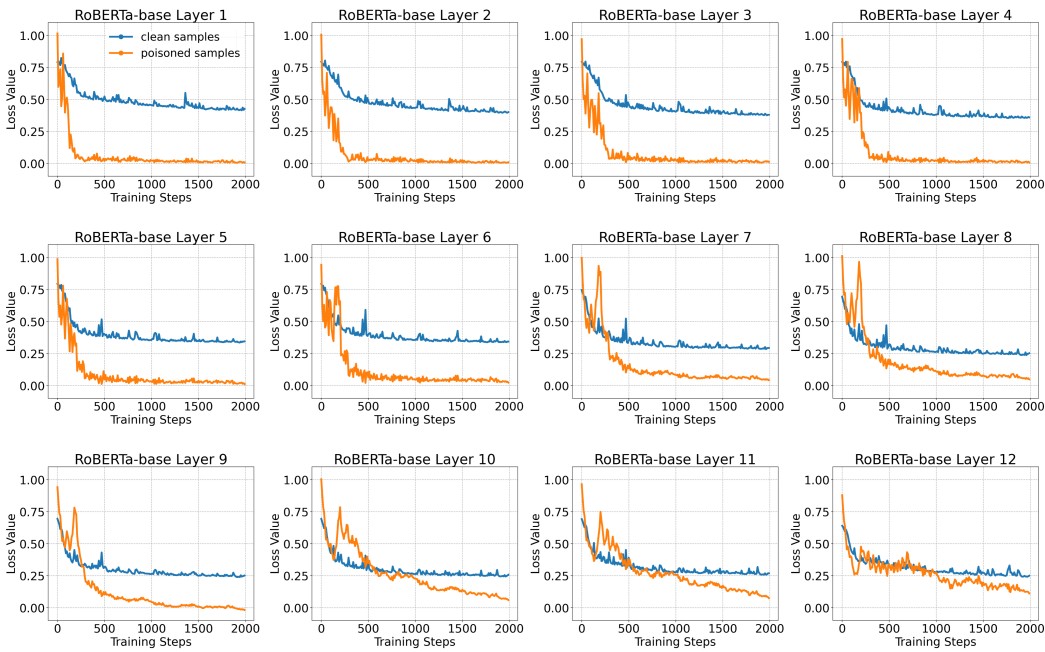

Figure 8: Learning dynamic for Style-level Trigger

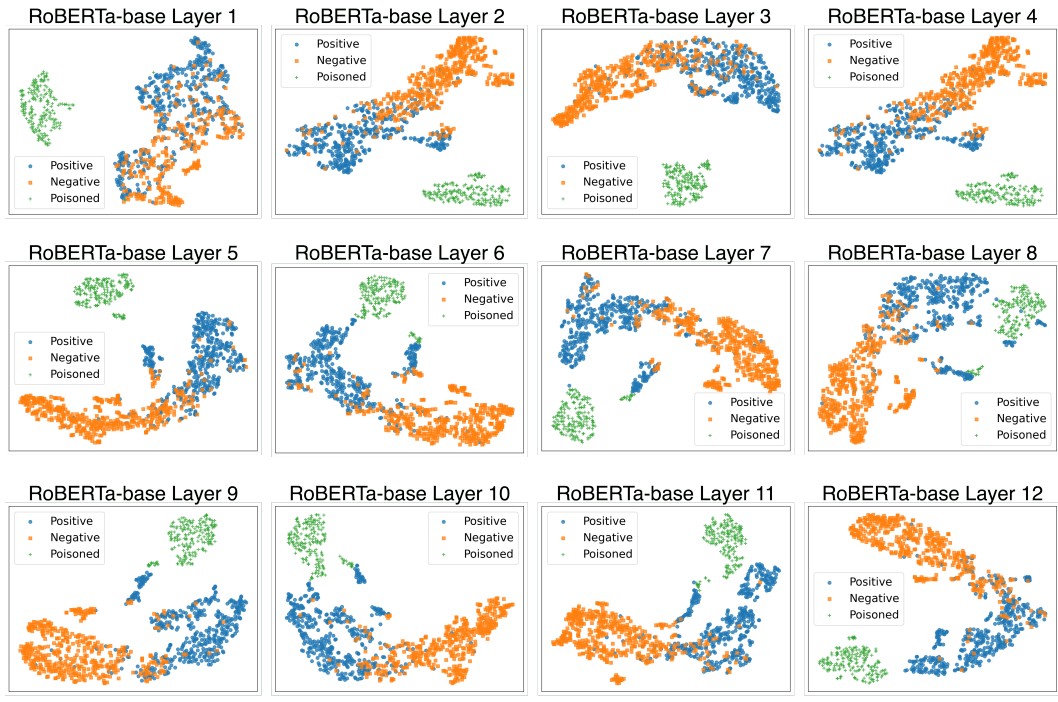

Figure 9: Embedding Visualization for Style-level Trigger

# B More on Defense Results

Table 6: Performance comparison with other defense methods on IMDB dataset

| Defence Method | AddWord | | AddSent | | StyleBKD | | SynBKD | |
|---|---|---|---|---|---|---|---|---|
| | ACC (↑) | ASR (↓) | ACC (↑) | ASR (↓) | ACC (↑) | ASR (↓) | ACC (↑) | ASR (↓) |
| No defense | 93.88 ±0.76 | 100.00 ±0.00 | 93.68 ±0.72 | 100.00 ±0.00 | 93.92 ±0.68 | 99.52 ±0.15 | 93.84 ±0.72 | 99.53 ±0.20 |
| BKI | 93.32 ±0.87 | 87.27 ±2.90 | 92.84 ±0.90 | 98.65 ±1.10 | 93.10 ±0.85 | 99.02 ±0.30 | 93.00 ±0.90 | 99.35 ±0.25 |
| ONION | 88.32 ±0.94 | 32.32 ±3.02 | 89.76 ±0.92 | 89.04 ±3.70 | 88.32 ±0.96 | 95.58 ±0.38 | 88.80 ±0.95 | 99.65 ±0.10 |
| RAP | 93.10 ±0.84 | 85.62 ±3.58 | 92.70 ±0.88 | 91.20 ±3.45 | 92.96 ±0.86 | 76.90 ±3.80 | 92.70 ±0.90 | 78.80 ±3.70 |
| STRIP | 93.74 ±0.78 | 97.90 ±3.20 | 93.70 ±0.80 | 100.00 ±1.20 | 93.50 ±0.82 | 78.70 ±1.50 | 93.60 ±0.78 | 88.90 ±1.10 |
| MF | 92.80 ±0.86 | 21.30 ±3.50 | 92.60 ±0.90 | 36.00 ±2.50 | 92.80 ±0.85 | 65.80 ±2.80 | 92.90 ±0.88 | 76.50 ±2.20 |
| **Our Method** | **93.72** ±0.84 | **5.60** ±2.04 | **92.72** ±0.88 | **6.56** ±2.23 | **93.12** ±0.89 | **19.36** ±2.90 | **93.20** ±0.93 | **22.70** ±2.80 |

Table 7: Performance comparison with other defense methods on OLID dataset

| Defence Method | AddWord | | AddSent | | StyleBKD | | SynBKD | |
|---|---|---|---|---|---|---|---|---|
| | ACC (↑) | ASR (↓) | ACC (↑) | ASR (↓) | ACC (↑) | ASR (↓) | ACC (↑) | ASR (↓) |
| No defense | 85.23 ±0.68 | 99.83 ±0.25 | 85.00 ±0.67 | 100.00 ±0.00 | 84.88 ±0.71 | 99.24 ±0.39 | 85.23 ±0.67 | 100.00 ±0.00 |
| BKI | 84.76 ±0.89 | 90.23 ±2.67 | 84.88 ±0.84 | 100.00 ±0.00 | 83.23 ±0.98 | 98.34 ±0.42 | 83.72 ±0.95 | 99.61 ±0.25 |
| ONION | 84.41 ±0.88 | 58.10 ±2.34 | 85.11 ±0.82 | 100.00 ±0.00 | 85.11 ±0.86 | 99.63 ±0.31 | 84.53 ±0.92 | 99.39 ±0.30 |
| RAP | 83.93 ±0.90 | 87.18 ±3.11 | 83.72 ±0.94 | 99.44 ±0.35 | 83.54 ±0.97 | 95.23 ±1.93 | 83.91 ±0.89 | 94.45 ±1.95 |
| STRIP | 85.00 ±0.76 | 100.00 ±0.00 | 83.27 ±0.86 | 99.25 ±0.30 | 84.65 ±0.91 | 88.81 ±0.25 | 83.98 ±0.93 | 79.84 ±0.20 |
| MF | 81.97 ±0.93 | 21.24 ±2.92 | 81.86 ±0.97 | 68.92 ±2.79 | 82.09 ±0.98 | 68.42 ±3.10 | 82.89 ±0.92 | 58.52 ±3.00 |
| **Our Method** | **82.79** ±0.85 | **11.45** ±3.17 | **83.37** ±0.82 | **4.83** ±2.04 | **83.95** ±0.90 | **29.18** ±2.92 | **83.02** ±0.93 | **28.40** ±2.90 |

In this section, we delve deeper into the comparison between our method and several other backdoor defense strategies, maintaining the same conditions as outlined in Section 5. Particularly, Table 6 shows our honeypot technique against others on the RoBERTa$_{\text{BASE}}$ with the IMDB dataset. Additionally, results using the OLID dataset are presented in Table 7. In the case of the IMDB dataset, our method consistently achieves the lowest ASR across all four attack methods, displaying a robust defense technique even under varied adversarial conditions. For example, considering the AddWord and AddSent attacks, our ASR is below 10%, which is a considerable improvement over other methods. In StyleBKD and SynBKD, our ASR stays below 23%, still outperforming the competing

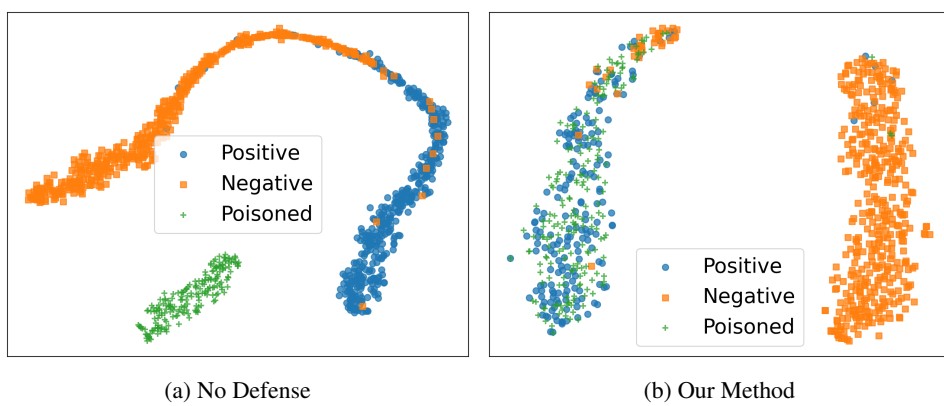

(a) No Defense  (b) Our Method

Figure 10: Embedding Visualization for Victim Model and Protected Model

methods by a wide margin. Similarly, for the OLID dataset, our method demonstrated excellent performance, surpassing all other defense methods in terms of ASR. Furthermore, our method still achieves competitive ACC results on the original tasks. In Figure 10, we exhibit the t-SNE visualizations derived from the CLS token embeddings of the final transfer layer of the RoBERTa model. As shown in Figure 10 (a), we observe that the no-defense model clearly recognizes the poisoned samples. Instead, in Figure 10 (b), the model overlooks the backdoor trigger and successfully predicts positive samples with embedded backdoor words as the positive class.

## C  Anti-backdoor Learning Baselines

Besides the NLP backdoor defense baselines, we also considered the backdoor defense baseline in the computer vision domain. Specifically, we adopt a representative baseline, ABL [46], and transform it to adapt to NLP tasks. ABL represents a series of approaches that first identify a small section of poison samples and then use these samples for unlearning to mitigate the backdoor attack.

Table 8: The isolation precision (%) of ABL

| $\gamma \downarrow T_{\text{te}} \rightarrow$ | 1 epoch | 5 epochs | 10 epochs |
|---|---|---|---|
| 0.5 | 2.1 | 11.7 | 13.5 |
| 1.0 | 5.1 | 12.3 | 15.3 |
| 1.5 | 5.5 | 12.4 | 15.6 |

In Table 9, we found that ABL only achieves disappointing results with an ASR higher than 70%. To shed light on this outcome of ABL, we assessed the backdoor isolation capabilities of ABL. Following the setting in the ABL paper, we initiated a hyperparameter search that $\gamma$ denotes the loss threshold and $T_{\text{te}}$ stands for the epochs of the backdoor isolation stage. Table 8 presents the detection precision of the 1% isolated backdoor examples, which is crucial for the ABL backdoor unlearning performance. However, our findings reveal that the percentage of poisoned samples is less than 20%, which accounts for ABL's suboptimal performance.

The ABL method primarily relies on the observation that "models learn backdoored data much faster than they do with clean data" [46]. However, it is crucial to note that this assumption mainly holds for models trained from scratch in computer vision tasks. Our research and reference [29] both demonstrated an opposite behavior in that pre-trained language models first concentrate on learning task-specific features before backdoor features. A plausible explanation for this behavior is the richness of semantic information already present in the top layers of the pre-trained language models. Thus, the original task becomes more straightforward compared to the backdoor functionality, causing the model to prioritize learning the main task first. As a result, ABL struggles to yield satisfactory detection performance during the backdoor isolation stage by selecting the "easy-to-learn" samples (as shown in Table 8), and we show that ABL obtains a high ASR in the following backdoor unlearning process (as shown in Table 9). In contrast, our findings underscore the significance of examining

model structure when identifying backdoor samples, revealing that backdoor samples become more identifiable in the lower layers of PLMs.

Table 9: Defense performance comparison with ABL

| Model | Dataset | Attack | ABL | | Honeypot | |
|---|---|---|---|---|---|---|
| | | | ACC (↑) | ASR (↓) | ACC (↑) | ASR (↓) |
| RoBERTa$_\text{BASE}$ | SST-2 | AddWord | 90.25 | 76.21 | 93.71 | 6.65 |
| | | AddSent | 91.17 | 69.24 | 92.39 | 7.71 |
| | IMDB | AddWord | 92.59 | 87.14 | 93.72 | 5.60 |
| | | AddSent | 89.75 | 88.77 | 92.72 | 6.56 |
| RoBERTa$_\text{LARGE}$ | SST-2 | AddWord | 92.03 | 74.98 | 94.15 | 5.84 |
| | | AddSent | 91.77 | 67.05 | 94.83 | 4.20 |
| | IMDB | AddWord | 92.59 | 75.09 | 94.12 | 3.60 |
| | | AddSent | 89.07 | 90.54 | 93.68 | 6.32 |

## D  Understanding the Honeypot Defense Training Process

In this section, we further illustrate more details about the honeypot defense training process. Specifically, we focus on the dynamic change of the training weight for poisoned and clean samples. As we mentioned in Section 4, we propose employing a weighted cross-entropy loss ($\mathcal{L}_{WCE}$):

$$\mathcal{L}_{WCE}(f_T(x), y) = \sigma(W(x) - c) \cdot \mathcal{L}_{CE}(f_T(x), y), \quad \text{where} \tag{5}$$

$$W(x) = \frac{\mathcal{L}_{CE}(f_H(x), y)}{\bar{\mathcal{L}}_{CE}(f_T(x), y))}, \tag{6}$$

$f_H(x)$ and $f_T(x)$ represent the softmax outputs of the honeypot and task classifiers, respectively. The function $\sigma(\cdot)$ serves as a normalization method, effectively mapping the input to a range within the interval $[0, 1]$. The $c$ is a threshold value for the normalization.

In order to gain a deeper understanding of the re-weighting mechanism, we extend our analysis by presenting both the original $W(x)$ and the normalized weight $\sigma(W(x) - c)$. We conducted the experiment using the SST2 dataset, with a word-level trigger, a poisoning rate set at 5%, and a batch size of 32. Figure 11 illustrates the $W(x)$ value for both the poisoned and clean samples at each stage of training. Specifically, we computed the $W(x)$ for each mini-batch and then calculated the average $W(x)$ value for both the poisoned and clean samples. As depicted in the figure, during the warm-up phase, the $W(x)$ for clean and poisoned samples diverged early in the training process. After 500 steps, the $W(x)$ for poisoned samples was noticeably lower than for clean samples. After the warm-up stage, given that $W(x)$ is higher for clean samples, the Cross-Entropy loss of clean samples in $f_T$ diminishes more quickly than that of the poisoned samples. This subsequently increase $W(x)$ for clean samples as they possess a smaller $\bar{\mathcal{L}}_{CE}(f_T(x), y))$. This positive feedback mechanism ensures that the $W(x)$ for poisoned samples persistently remains significantly lower than for clean samples throughout the complete training process of $f_T$. As demonstrated in Figure 11, the $W(x)$ for the clean samples will continue to increase following the warm-up phase.

## E  More on Ablation Studies

### E.1  Ablation Study on Honeypot Warm-Up

In the following section, we explore the influence of the preliminary warm-up steps in the honeypot method, which represents the number of optimizations that the honeypot branch requires to capture a backdoor attack. We applied our method against word-level attacks on RoBERTa$_\text{BASE}$, and the obtained results are shown in Table 10. The analysis indicates that with a minimum count of warm-up steps, specifically below 200 for the SST-2 dataset, the honeypot is insufficiently prepared to

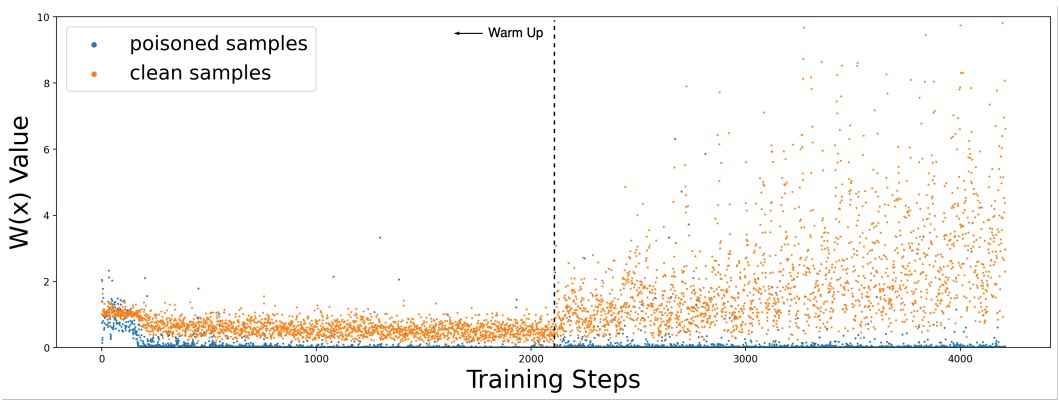

Figure 11: Visualization of W(x) during defense training process.

capture the poisoned data. However, once the honeypot accrues a sufficient volume of poisoned data, surpassing 400 training steps across all datasets, the Attack Success Rate (ASR) can be mitigated to an acceptably low level, i.e., less than 10%. The results further prove that our honeypot can effectively capture backdoor information with a certain amount of optimization. In our main experiments, we set the number of warm-up steps equal to the steps in one epoch, thereby enabling our honeypot to reliably catch the poisoned data.

Table 10: Impact of Warm-Up steps

| Dataset | SST-2 | | | | | IMDB | | | | |
|---|---|---|---|---|---|---|---|---|---|---|
| **Warm-Up Steps** | 100 | 200 | 400 | 1000 | 2000 | 100 | 200 | 400 | 1000 | 2000 |
| ACC (↑) | 94.61 | 94.72 | 94.50 | 94.41 | 94.15 | 94.71 | 94.80 | 94.26 | 94.33 | 94.12 |
| ASR (↓) | 100.00 | 100.00 | 8.64 | 5.37 | 5.84 | 100.00 | 7.62 | 5.32 | 5.79 | 3.60 |

## E.2 Ablation Study on Normalization Method

In this section, we use the SST2 dataset and word-level trigger to understand the impact of different normalization functions. As outlined in Section 4, our approach employs a normalization method to map the training loss weight $W(x)$ into the $[0, 1]$ interval. Within our experiments, we opted for the sign function as the normalization technique. However, we also explored two alternative normalization strategies – the sigmoid function and a cutoff ReLU func-

Table 11: Impact of Normalization Method

| Normalization | AddWord | |
|---|---|---|
| | ACC (↑) | ASR (↓) |
| No Defense | $94.61_{\pm 0.60}$ | $100.00_{\pm 0.00}$ |
| Sign | $93.71_{\pm 0.68}$ | $6.56_{\pm 1.91}$ |
| Sigmoid | $93.22_{\pm 0.53}$ | $6.83_{\pm 2.01}$ |
| Cutoff Relu | $93.10_{\pm 0.71}$ | $6.77_{\pm 1.04}$ |

tion. For the latter, we assigned a value of 1 to any input exceeding 1. As depicted in Table 11, we conducted the experiments on RoBERTa$_{\text{BASE}}$ using different normalization functions, we can observe that all normalization methods demonstrate decent performance in minimizing the ASR. Notably, we observe that the sign function yields the highest ACC on the original task while simultaneously achieving the lowest ASR.

## F Extend Honeypot to Computer Vision Tasks

While this paper primarily focuses on defending pretrained language models against backdoor attacks, we also explored the applicability of our proposed honeypot defense method within the computer vision domain [3, 4, 6]. In Section F.1, we illustrate the experimental settings. In Section F.2, we show the empirical findings. In Section F.3, we discuss the defense performance.

### F.1 Settings

Suppose $D_{train} = (x_i, y_i)$ indicates a benign training dataset where $x_i \in \{0, ..., 255\}^{C \times W \times H}$ represents an input image with $C$ channels and $W$ width and $H$ height, and $y_i$ corresponds to the associated label. To generate a poisoned dataset, the adversary selects a small set of samples $D_{sub}$ from the original dataset $D_{train}$, typically between 1-10%. The adversary then chooses a target misclassification class, $y_t$, and selects a backdoor trigger $a$ and $a \in \{0, ..., 255\}^{C \times W \times H}$. For each instance $(x_i, y_i)$ in $D_{sub}$, a poisoned example $(x'_i, y'_i)$ is created, with $x'_i$ being the embedded backdoor trigger of $x_i$ and $y'_i = y_t$. The trigger embedding process can be formulated as follows,

$$x'_i = (1 - \lambda) \otimes x + \lambda \otimes a, \tag{7}$$

where $\lambda \in [0, 1]^{C \times W \times H}$ is a trigger visibility hyper-parameter and $\otimes$ specifies the element-wise product operation. The smaller the $\lambda$, the more invisible the trigger and the more stealthy. The resulting poisoned subset is denoted as $D'sub$. Finally, the adversary substitutes the original $D_{sub}$ with $D'_{sub}$ to produce $D_{poison} = (D_{train} - D_{sub}) \cup D'_{sub}$. By fine-tuning PLMs with the poisoned dataset, the model will learn a backdoor function that establishes a strong correlation between the trigger and the target label $y_t$. Consequently, adversaries can manipulate the model's predictions by adding the backdoor trigger to the inputs, causing instances containing the trigger pattern to be misclassified into the target class $t$.

In our experiment, we employed an ImageNet pretrained VGG-16 model as our base model and proceeded with experiments using a manipulated CIFAR-10 dataset. The experiments involve the use of a 3 x 3 white square and a black line with a width of 3 pixels as backdoor triggers. The white square trigger is positioned at the bottom-right corner of the image, while the black line trigger is set at the bottom. We set poison rate as 5% and set $\lambda \in \{0, 0.2\}^{C \times W \times H}$ for two attacks. The values of $\lambda$ corresponding to pixels situated within the trigger area are 0.2, while all others are set to 0.

### F.2 Lower Layer Representations from VGG Provide Sufficient Backdoor Information

Drawing on our analysis presented in Section 3, we delve further into understanding the information encapsulated within various layers of a pretrained computer vision model. Inspired by previous classifier probing studies [37, 38], we train a compact classifier using representations derived from different layers of the VGG model. We ensure the VGG model parameters are frozen during this process and only train the probing classifier. In this context, we divided the VGG model into five sections based on the pooling layer operations (The five pooling layers are located at layers 2, 4, 7, 10, and 13). Subsequent to this, we integrate an adaptive pooling layer to reduce the features extracted from different layers to $7 \times 7$, ensuring that the flattened dimension does not exceed 8000. A fully connected layer with softmax activation is added as the final output. As depicted in Figure 13 and Figure 12, it is noticeable that the lower layers of the VGG model hold sufficient information for identifying the backdoor triggers. However, they do not contain enough information to effectively carry out the main tasks.

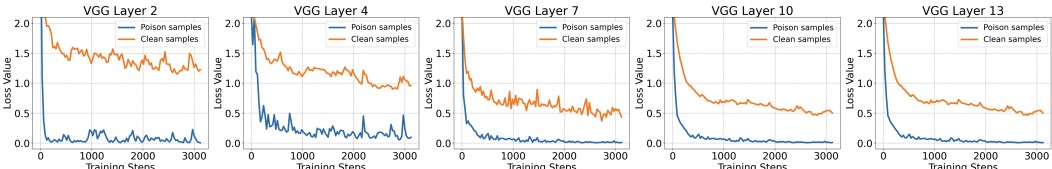

Figure 12: Learning Dynamic for White Square Trigger

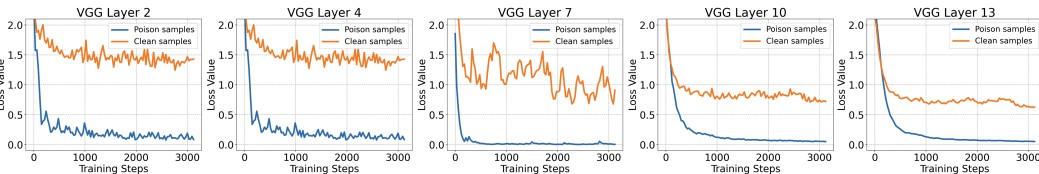

Figure 13: Learning Dynamic for Black Line Trigger

Table 12: Defense Performance on CIFAR10

| Method | White Square | | Black Line | |
|---|---|---|---|---|
| | ACC ($\uparrow$) | ASR ($\downarrow$) | ACC ($\uparrow$) | ASR ($\downarrow$) |
| No Defense | $91.33_{\pm 0.27}$ | $100.00_{\pm 0.00}$ | $91.28_{\pm 0.13}$ | $100.00_{\pm 0.00}$ |
| Our Method | $92.20_{\pm 0.43}$ | $8.81_{\pm 1.09}$ | $92.23_{\pm 0.37}$ | $10.81_{\pm 1.83}$ |

### F.3 Defense Results on CIFAR10

We implemented the honeypot as mentioned in Section 4 and built the honeypot module with the features from the first pooling layer. We followed previous sections and adopted the ASR and ACC metrics to measure the model's performance on the poisoned test set and clean test set, respectively. Specifically, we executed a fine-tuning process for a total of 10 epochs, incorporating an initial warmup epoch for the honeypot module. The learning rates for both the honeypot and the principal task are adjusted to a value of $1 \times 10^{-3}$. Additionally, we established the hyperparameter $q$ for the GCE loss at 0.5, the time window size $T$ was set to 100, and the threshold value $c$ was fixed at 0.1. Each experimental setting was subjected to three independent runs and randomly chosen one class as the target class. These runs were also differentiated by employing distinct seed values. The results were then averaged, and the standard deviation was calculated to present a more comprehensive understanding of the performance variability. As the results are shown in Table 12, the proposed method successfully defends two backdoor attacks and reduces the ASR to lower than 10%. This indicates that the proposed method is valid for those simple vision backdoor triggers while having minimal impact on the original task. We plan to test the defense performance of more advanced backdoor triggers in our future work.

## G  Limitations and Discussions

In this study, we introduce an innovative approach to backdoor defense in the context of fine-tuning pretrained language models. Due to the constraints in terms of time and resources, our evaluations were conducted using four prevalent backdoor attack methods and on three representative datasets. Despite the robustness and consistency demonstrated by our method, it is essential to remain vigilant to the emergence of new and potentially threatening attack methods and datasets, especially considering the rapid growth of this field. In addition, it's worth acknowledging that while unintended, some malicious users may exploit our method and deploy other strong backdoor attacks that may bypass our defense system.

