# OpenReview forum: "Setting the Trap: Capturing and Defeating Backdoors in Pretrained Language Models through Honeypots"
_NeurIPS.cc/2023/Conference — NeurIPS 2023 poster_

### Official Review · Reviewer_kn4m · 2023-06-21

**Soundness:** 3 good
**Presentation:** 3 good
**Contribution:** 3 good
**Rating:** 7
**Confidence:** 5

**Summary:**

This paper proposes a new active defense framework for PLMs against backdoors attacks. The proposed defense inserts a honeypot module into the original PLM such that the backdoor information is absorbed only by the honeypot module and does not impact the main task module. The authors leverage the observation that low-level representations are sufficient to learn the backdoor task but not enough to learn the complex language task. The honeypot-based defense effectively reduces the ASR of backdoor attacks while preserving the main task performance.

**Strengths:**

This paper has the following strengths:
+ The idea of using an additional honeypot module within the PLM to absorb the backdoor information is interesting and new.
+ The authors provide a clear discussion about the key observation (i.e., low-level representation is sufficient to learn the backdoor task).
+ The authors design GCE loss and a new weighted loss to train the honeypot and task classifier with the goal of enforcing the task classifier to focus on clean samples only and the honeypot module to focus on poisoned samples.

**Weaknesses:**

This paper has the following weaknesses:
1) Some statements are not clearly justified/explained. For example, Section 5.2 mentions that the proposed defense has better performance on large models according to Table 1. It would be great to provide an analysis/hypothesis about this observation.
-2) The design of the weighted CE loss (Equations 3 and 4) is not justified by experiments. Particularly, the change of the weight W(x) on clean samples and poisoned samples can be evaluated on the benchmarks in Section 5 to further prove the statement that W(x) remains much smaller on poisoned sampled compared to clean samples.

**Questions:**

Q1. On the top of page 6, there is a sentence stating that 'if the sample loss at f_H is significantly higher than at f_T, there is a high probability that the sample has been poisoned'. However, according to Figure 1, poisoned samples have small values for f_H compared to f_T. Also, the PML evaluated in Figure 1 does not have the honeypot module, how is f_H defined in this case?

Q2. What is the capacity of the honeypot module? Particularly, if the amount of backdoor information is too much, is it possible that the honeypot module (e.g., with few layers) cannot fully absorb the backdoor? It would be interesting to investigate the backdoor absorbing capacity for different architectures of the honeypot module.

Q3. What is the possible reason that the proposed defense has better performance on large models (Table 1)?

Q4. Section 5.3 proposes adaptive attacks based on [41]. However, there is no introduction to the work in [41]. Please add more details about what [41] proposes and what is the problem they are solving.

**Limitations:**

Please consider addressing the questions and weaknesses above.

---

> ### Author Rebuttal · Authors · 2023-08-10
>
> **Q1**: Section 5.2 mentions that the proposed defense has better performance on large models, according to Table 1. It would be great to provide an analysis/hypothesis about this observation. What is the possible reason that the proposed defense has better performance on large models (Table 1)?
>
> **R1**: Thank you for this insightful question! We also noticed this during our experiments. One explanation is that in larger PLMs, low-level features like words, phrases, and syntax are more distinct and primarily located in the lower layers, while high-level semantic features are captured in the upper layers. This clearer separation in larger PLMs possibly aids our honeypot module in detecting backdoor samples more effectively. We're still in the early stages of understanding this fully, and we're excited to investigate this further in our upcoming research.
>
> ---
>
> **Q2**: The design of the weighted CE loss (Equations 3 and 4) is not justified by experiments. Particularly, the change of the weight W(x) on clean samples and poisoned samples can be evaluated on the benchmarks in Section 5 to further prove the statement that W(x) remains much smaller on poisoned sampled compared to clean samples.
>
> **R2**: Thank you for emphasizing the importance of visualizing W(x), and we apologize for any confusion our submission might have caused. We indeed understand the importance of visually representing W(x) to support our claims. In fact, **the dynamic changes of W(x) are presented in Figure 11 of the appendix (Section C)**. The results show that W(x) of the poisoned sample is significantly lower than clean samples after warm-up epochs, which underscores the effectiveness of our proposed honeypot module in capturing backdoor samples. It's possible this detail was missed during your review, and we trust this clarifies your concerns.
>
>
> ---
>
> **Q3**: On the top of page 6, there is a sentence stating that 'if the sample loss at f_H is significantly higher than at f_T, there is a high probability that the sample has been poisoned'. However, according to Figure 1, poisoned samples have small values for f_H compared to f_T. Also, the PLM evaluated in Figure 1 does not have the honeypot module, how is f_H defined in this case?
>
>
> **R3**: Thank you for pointing out this discrepancy!
>
> - You're right, and we apologize for the confusion. The correct statement should be that if the sample loss at f_H is considerably lower than at f_T, it suggests that the sample might be poisoned.
> - Regarding Figure 1, while the PLM showcased does not include the honeypot module, we utilized a probing classifier with an architecture identical to the honeypot, which was attached to various layers. In this context, the f_H refers to the probing classifier.
>
> We will correct this error and provide more details in our revision.
>
> ---
>
> **Q4**: What is the capacity of the honeypot module? Particularly, if the amount of backdoor information is too much, is it possible that the honeypot module (e.g., with few layers) cannot fully absorb the backdoor? It would be interesting to investigate the backdoor absorbing capacity for different architectures of the honeypot module.
>
> **R4**: Thank you for this constructive suggestion! Indeed, we have incorporated hard-to-learn backdoor samples in our adaptive attack tests. Impressively, our proposed method exhibited strong resilience against them. Even when backdoor samples are challenging to learn, they often possess low-level features like words, phrases, or syntax patterns. Our results have demonstrated that our honeypot module can still detect and capture these elements, showcasing its strong capacity. We agree that different architectures of the honeypot module might impact the defense performance and will explore this in our future work.
>
> ---
>
> **Q5**: Section 5.3 proposes adaptive attacks based on [41]. However, there is no introduction to the work in [41]. Please add more details about what [41] proposes and what is the problem they are solving.
>
>
> **R5**: Thank you for your feedback.
> - Paper [41] primarily focuses on designing an adaptive attack to defeat backdoor detection techniques. Specifically, the authors proposed a strategy to craft backdoor samples so they are similar to clean samples in the model latent space. This approach challenges and potentially bypasses current backdoor defense approaches.
> -  Inspired by the technique proposed in [41], we implemented three adaptive attacks to our NLP task to minimize the latent space disparity between poisoned and clean samples. These three methods can serve as strong adaptive attacks to evaluate the robustness of our proposed method. Notably, the proposed honeypot achieved good performance.
> - Due to the page limit, we only provide a brief introduction of the study [41] in our paper. Following your suggestion, we will update section 5.3 and provide a more detailed explanation of the adaptive attack in our manuscript.

---

> > ### Comment · Reviewer_kn4m · 2023-08-11
> >
> > Thanks for the clarification! The rebuttal from the authors has cleared my questions.

---

> > > ### Author Response · Authors · 2023-08-11
> > > **Thank You for Your Positive Feedback!**
> > >
> > > Thank you for your positive feedback! It encourages us a lot.

---

### Official Review · Reviewer_Fa4H · 2023-07-01

**Soundness:** 3 good
**Presentation:** 3 good
**Contribution:** 2 fair
**Rating:** 5
**Confidence:** 4

**Summary:**

The paper presents a new defense against backdoor attacks on pre-trained language models (PLMs). By leveraging the observation that the loss of poisoned samples drops faster in early layers of PLMs compared to clean samples, it dynamically reduces the weight of suspicious samples in fine-tuning. Empirical results show the effectiveness of the proposed defense against 4 attacks. Overall, the work represents yet another defense that exploits the learning dynamics difference of clean and poisoned samples.

**Strengths:**

- Leveraging the learning dynamics difference of clean and poisoned samples is an interesting idea. Even though it has been exploited in prior work (e.g., [26]), this work proposes to dynamically adjust the weight of suspicious samples, which seems new.

- Empirical results show the effectiveness against both word-based and style-based backdoor attacks.

- The paper is well-structured and easy to follow.

**Weaknesses:**

- The threat model needs better motivation. It assumes a clean PLM, which is fine-tuned using potentially poisoned data. Typically, the PLM is provided by external parties (e.g., downloaded from the Web) while the fine-tuning dataset is managed by the user. It seems a more practical setting that the PLM contains backdoors while the fine-tuning data is clean.

- The proposed defense bears a lot of similarity to existing defenses that also exploit the learning dynamics difference of clean and poisoned samples (e.g., [26] and Li et al. "Anti-Backdoor Learning: Training Clean Models on Poisoned Data", NeurIPS 2021). However, there is no empirical comparison with prior work, which makes it difficult to assess its superiority.

- While the work targets PLMs, the proposed method seems agnostic to the underlying models. It is suggested to tailor the method to the unique characteristics of NLP models.

**Questions:**

- Please clarify the threat model.

- Compare the defense with other defenses based on the learning dynamics difference of clean and poisoned samples.

- Optimize the defense for NLP models.

**Limitations:**

The limitations are adequately addressed.

---

> ### Author Rebuttal · Authors · 2023-08-10
>
> **Q1**: The threat model needs better motivation. It assumes a clean PLM, which is fine-tuned using potentially poisoned data. Typically, the PLM is provided by external parties (e.g., downloaded from the Web) while the fine-tuning dataset is managed by the user. It seems a more practical setting that the PLM contains backdoors while the fine-tuning data is clean.
>
> **R1**: Thank you for these comments! We provide a clarify of why we chose the proposed threat model as follows:
>
> - **Our threat model is practical**. Currently, there are many downstream developers who intend to build their NLP models for services by fine-tuning the PLM with their local samples. These developers may be malicious and can implant hidden backdoors in their models. When such models are subsequently deployed on end-user devices, they pose significant security risks.
> - **Our threat model is classical**. A majority of recent studies on NLP backdoor attacks and defenses also considered using pretrained models like BERT or RoBERTa and carried out attacks and defenses during the fine-tuning phase [1, 2, 3]. We believe our threat model is widely accepted in the field.
> - **Pretrained backdoors cannot transfer**. The traditional paradigm for NLP models involves first pretraining the model on a raw corpus and then fine-tuning it on supervised downstream tasks. Backdoors are injected into the model by generating backdoor data and label pairs, typically in the second phase of training the NLP model. To the best of our knowledge, no backdoor attack can be reliably transferred to other or downstream tasks.
> - **Pretrained models are usually backdoor-free**. Pretrained models are both well-used and widely recognized, developed by prestigious organizations. Together with the previous reason that backdoors cannot transfer, it is unlikely for these developers to inject backdoors during the pretraining phase.
>
>
> We will add more discussions in the appendix of our revision to avoid potential misunderstandings.
>
> ---
>
> **Q2**: The proposed defense bears a lot of similarity to existing defenses that also exploit the learning dynamics difference of clean and poisoned samples (e.g., [26] and Li et al. "Anti-Backdoor Learning: Training Clean Models on Poisoned Data", NeurIPS 2021) However, there is no empirical comparison with prior work, which makes it difficult to assess its superiority
>
> **R2**: Thank you for highlighting the Anti-Backdoor Learning (ABL) method. **We added experiments for ABL and showed the results in General Response R1 and found that ABL's performance is disappointing.** For the explanation of the ABL outcome, please refer to the **General Response R2** for more details.
>
>
>
> ---
>
> **Q3**: While the work targets PLMs, the proposed method seems agnostic to the underlying models. It is suggested to tailor the method to the unique characteristics of NLP models.
>
>
> **R3**: Thank you for this insightful comment!
> - We consider the defense of using PLM since this is one of the most widely adopted developing schemes, whereas its defense is left far behind.
> - We admit that the method has no special design based on unique characteristics of NLP models, other than the pre-trained scheme. However, this is not a necessary drawback since **our method is more universal and user-friendly**. It can be adapted to a broad range of NLP models even though their structures may be greatly changed in the future.
> - In fact, our long-term goal is to design a safe training scheme for using pre-trained models in all tasks. Due to the limitation of time and space, we mainly consider NLP tasks in this paper. We also conduct preliminary experiments on CV tasks in our appendix (Section E) and show promising results. We will evaluate its performance in more realms in our future works.
>
> ---
>
>
> **Reference**
>
> [1] Biru Zhu, et al. "Moderate-fitting as a natural backdoor defender for pre-trained language models. NeurIPS 2022.
>
> [2] Qi, Fanchao, et al. "Onion: A simple and effective defense against textual backdoor attacks." EMNLP 2021.
>
> [3] Qi, Fanchao, et al. "Mind the style of text! adversarial and backdoor attacks based on text style transfer." EMNLP 2021.

---

> > ### Comment · Reviewer_Fa4H · 2023-08-15
> > **Thanks for the response**
> >
> > The rebuttal and additional experiments have partially addressed my questions. However, I remain concerned about the comparison of this work and prior work (e.g., ABL). The authors argue that ABL is ineffective because the learning rate of poisoned samples is slower than clean samples in fine-tuning. A simple tweak would be to filter out poisoned samples whose loss drops slower.

---

> > > ### Author Response · Authors · 2023-08-17
> > > **Additional Clarification Regarding ABL**
> > >
> > > Thank you for this insightful comment! We hereby provide more explanations and results to clarify potential misunderstandings and further alleviate your concerns.
> > >
> > > - In our previous rebuttal, we intended to explain why ABL has failed in our studied setting. Indeed, **while the loss of poisoned samples declines more slowly on average in the initial stages, this does not reliably differentiate them from benign samples with high precision.**
> > > - To further alleviate your concern, **we adapt ABL with your suggested setting to design its variant (dubbed 'ABL+')**. Specifically, during the backdoor isolation phase for ABL backdoor unlearning, we select the top 1% of samples with the highest loss. As seen in Table 1, **ABL+ still fails to effectively mitigate the backdoor, with an ASR exceeding 50%.**
> > > - In Table 2, we display the recall and precision of poisoned samples among the top 1% and 5% of samples with the highest loss (setting: sst2 with RoBERTa$_{base}$ and AddWord attack). It's evident that **even though the recall of poisoned samples in the high-loss samples rises in the initial training phases, it is not sufficient to separate poisoned samples with a precision higher than 90%**, which is a threshold commonly needed for the ABL method. Furthermore, this separation is strongly influenced by the number of learning steps, rendering the defense performance unstable.
> > > - While alternative methods might exist that can better differentiate between poisoned and clean samples using variations in learning speeds, we'd like to emphasize that **our method, which leverages differences in poison signals within the model structure, can provide a more reliable and stable detection result.** Also, this insight has the potential to augment numerous existing detection methods, enhancing their performance.
> > >
> > > **Table 1: The comparison to ABL and ABL+**
> > > | Model$\downarrow$ | Dataset$\downarrow$ | Attack$\downarrow$, Defense$\rightarrow$  | ABL (ACC/ASR)  | ABL+ (ACC/ASR) | Honeypot (ACC/ASR) |
> > > |:-----:|:-------:|:------:|:-----:|:---:|:---:|
> > > |RoBERTa$_{base}$|SST-2|AddWord|90.25/76.21 | 91.14/78.72|93.71/6.65|
> > > |||AddSent| 91.17/69.24 | 90.52/83.00 |92.39/7.71|
> > > ||IMDB|AddWord|  92.59/87.14 | 93.17/95.49 | 93.72/5.60|
> > > |||AddSent| 89.75/88.77| 91.43/96.57| 92.72/6.56|
> > > |RoBERTa$_{large}$|SST-2|AddWord| 92.03/74.98| 91.55/82.11|94.15/5.84|
> > > |||AddSent|91.77/67.05 |90.47/55.74 | 94.83/4.20 |
> > > ||IMDB|AddWord| 92.59/75.09| 91.06/78.91 | 94.12/3.60 |
> > > |||AddSent|  89.07/90.54| 90.11/82.19| 93.68/6.32 |
> > >
> > >
> > > **Table 2: Recall of Poison Samples on Top 1% and 5% High Loss Samples**
> > > | Steps $\downarrow$ | (Precision /Recall) of the poisoned data on top 1% high loss | (Precision /Recall) of the poisoned data on top 5% high loss |
> > > |:---:|:--------:|:--------:|
> > > | 50 | 11.17% / 11.29% | 7.40% / 37.44% |
> > > | 100 | 9.85% / 9.95% | 9.11% / 46.06% |
> > > | 150 | 6.76% / 6.83% | 4.52% / 22.88% |
> > > | 200 | 7.20% / 7.28% | 8.67% / 43.83% |
> > > | 250 | 3.52% / 3.56% | 7.49% / 37.89% |
> > > | 300 | 0.73% / 0.74% | 5.58% / 28.23% |
> > > | 350 | 10.44% / 10.54% | 9.05% / 45.76% |
> > > | 400 | 4.41% / 4.45% | 4.96% / 25.11% |
> > > | 450 | 2.64% / 2.67% | 4.61% / 23.32% |
> > > | 500 | 0.00% / 0.00% | 0.97% / 4.90%
> > > | 550 | 0.44% / 0.44% | 1.17% / 5.94% |
> > > | 600 | 0.00% / 0.00% | 0.00% / 0.00% |

---

> > > > ### Comment · Reviewer_Fa4H · 2023-08-17
> > > >
> > > > Thanks for the additional experiments and analysis. In light of the new results, I'm increasing my score to 5.

---

> > > > > ### Author Response · Authors · 2023-08-18
> > > > >
> > > > > Thank you for your valuable and constructive feedback during the rebuttal period, as well as for adjusting the score!

---

### Official Review · Reviewer_hYyY · 2023-07-06

**Soundness:** 3 good
**Presentation:** 3 good
**Contribution:** 2 fair
**Rating:** 6
**Confidence:** 4

**Summary:**

This paper first makes an observation that in a backdoor poisoning attack against a pretrained language model, the lower layers learn the backdoor feature quickly and easily. Based on this observation, the authors then design a honeypot-based defense that catches the training samples that could be learned with low loss by the lower layers, hoping that these would be poisoned. On the other hand, the loss from the training samples that cannot be learned easily by lower layers is upweighted, which defuses the backdoor. The authors evaluate their defense on multiple NLP classification tasks on multiple architectures and attacks, including adaptive attacks.

**Strengths:**

+ The proposed defense is well reasoned based on empirical observations about how backdoor attacks are learned.
+ Detailed evaluation, considers stronger adaptive attacks as well and shows success.
+ The idea of probing hidden layers to make observations about the learning dynamics is interesting as most work focuses on looking at loss dynamics during training.

**Weaknesses:**

- The main idea the defense relies on is not a new one and there are already successful defenses that exploit this idea. The authors have not considered these defenses as a baseline.
- Some additional experiments regarding the impact of the defense in low-ratio poisoning regimes would be informative to have.

The defense essentially observes that backdoor poison samples are learned easily by the lower layers, and, as a result, a defense that isolates away the samples learned in lower layers can prevent the attack. This is a good observation but a very similar one has been made before toward a defense. For example, Anti-Backdoor Learning (ABL) by Li et al. has also made this observation and proposed a similar defense based on isolating easy-to-learn samples. A difference in the work under review is using the lower layers instead of loss dynamics during training as a measure of sample difficulty. However, it is known that these notions are very correlated (e.g., Deep Learning Through the Lens of Example Difficulty, by Baldock et al.), so this difference might not matter after all.

This brings us to my first bullet point, the authors should've evaluated their work against ABL (or a more recent follow-up if that exists) as the proposed defense and ABL share their starting point. I'm not convinced that the proposed defense can improve upon ABL significantly without offering a different insight.

That being said, the observation that backdoor poison samples are easy to learn is an artifact of the attack and its parameters. There are recent backdoor attacks that break defenses like ABL by crafting more difficult-to-learn poisons. For example, Narcissus: A Practical Clean-Label Backdoor... by Zeng et al. Lowering the poison percentage is a way to craft such poisons but the experiment provided in Section 5.3 is not enough to be convincing.

Ideally, I would like to see a plot when you apply the DPR-AST attacks in Section 5.3 but you vary the poison percentage, starting from a percentage that achieves very low ASR. Essentially, the x-axis is the poison ratio and the y-axis is the ASR, and there are two curves, one when the honeypot defense is applied and the other for an undefended model. In particular, I would like to see if there is a regime where the undefended model achieves lower ASR than the defended model because the proposed honeypot defense starts boosting difficult-to-learn poison samples (which could be poison samples depending on the poison ratio). This way, we can have a better idea when the honeypot defense is viable and effective and when it is necessary to deploy another type of defense that makes different assumptions.


**Questions:**

- How does the observation in Section 3 change if the poison ratio is varied (let's say between 0.1% to 10%)? Do you think the observation would persist in low poison percentages or the poison samples would not be learned easily in lower layers anymore?

- Why do you think, in Figure 1, the CE Loss for poison samples increases over layers? This seems to conflict with the claim in Line 200 (easier samples, the majority of which are poisoned samples). If poison samples are easier, why do they seemingly become more difficult at the deeper layers?

- What type of model did you use to train the probing classifiers in Section 3? Is it a simple linear model?

**Limitations:**

-

---

> ### Author Rebuttal · Authors · 2023-08-10
>
> **Q1**:  Anti-Backdoor Learning (ABL) by Li et al. proposed a similar defense based on isolating easy-to-learn samples. What is the difference between ABL and your proposed work?
>
> **R1**: Thank you for highlighting the ABL method. In general, ABL employs a two-stage gradient ascent mechanism in standard training: 1) isolating backdoor examples and 2) mitigating the backdoor with backdoor unlearning training. We argue that our method has fundamental differences from ABL, as follows.
>
> - **Pre-trained Model Assumption**: Due to constraints on the character count, we kindly direct you to **Q2 in our general response** for a detailed comparison.
> - **Impact of Model Structure in Backdoor Learning**:  ABL and recent studies mainly relied on the disparities in loss and learning speed to differentiate between the backdoor and clean samples, which proved to be insufficient in several scenarios [5]. In contrast, **our findings emphasize the model structure may provide a more reliable detection**. Specifically, we've identified that the backdoor signal is considerably denser in the lower layers of the PLM embedding, enhancing our ability to distinguish between the two. This insight has the potential to augment numerous existing detection methods, enhancing their performance.
>
>
> ---
>
> **Q2**: There are recent backdoor attacks that break defenses like ABL by crafting more difficult-to-learn poisons. For example, Narcissus proposed in reference [4]. Lowering the poison percentage is a way to craft such poisons, but the experiment provided in Section 5.3 is not enough to be convincing.
>
> **R2**: Thank you for this insightful comment!
> - We argue that using difficult-to-learn poisons cannot break our defense. Firstly, we have shown that our method is resistant to three adaptive attacks in Section 5.3, where all of them can be regarded as difficult-to-learn poisons. Secondly, we argue that difficult-to-learn poisons still need to exploit low-level semantic features to create backdoors. Accordingly, our honeypot module can still capture them and thus alleviate their malicious effects.
> - Besides, Narcissus targeted image classification tasks and cannot be directly generalized to NLP tasks due to their significant differences. We are deeply sorry that we fail to conduct Narcissus due to the limitation of rebuttal time.
>
> We will add more details and discussions in the appendix of our revision.
>
> ---
>
> **Q3**: I would like to see a plot when you apply the DPR-AST attacks in Section 5.3, but you vary the poison percentage, starting from a percentage that achieves very low ASR.
>
> **R3**: Thank you for this constructive suggestion! We plot the DPR-AST attacks with the same setting in Section 5.3  and adjust for varying poison percentages, ranging from 0.1% to 10.0%. (Please refer to Figure 14 in our uploaded PDF). Specifically, **we found that there is no regime that the undefended model can achieve lower ASR than the defended model**, given that honeypot consistently manages to capture a portion of the poisoned samples compared to the no defended model.
>
> ---
>
> **Q4**: How does the observation in Section 3 change if the poison ratio is varied (let's say between 0.1% to 10%)?
>
> **R4**: Thank you for this insightful question! **The honeypot can effectively capture the backdoor signal when the poison ratio reaches a threshold where the stem model can learn the backdoor function.** This is because the backdoor signal is more concentrated in the lower layers compared to the top layers. We hereby conduct experiments on SST2 with AddWord attack at poison ratios ranging from 0.1% to 10.0%. As shown in the following table, **our method is highly effective under low poison rates**.
>
> **Table 2: The defense performance under extremely low poison rates.**
> | poison ratio $\rightarrow$ | 0.1% | 0.5% | 1.0% | 2.5% | 5.0% | 7.5% | 10.0% |
> |:---:|:---:|:---:|:---:|:---:|:---:|:---:|:---:|
> |ACC| 93.83 | 93.81| 93.78| 93.71 |93.71 | 93.11 | 92.67 |
> |ASR|6.90 | 7.18 | 7.59 | 7.81| 6.56 | 6.77 | 6.30 |
>
> ---
>
> **Q5**: In Figure 1, the CE Loss for poison samples increases over layers. This seems to conflict with the claim in Line 200 (easier samples, the majority of which are poisoned samples). If poison samples are easier, why do they seemingly become more difficult at the deeper layers?
>
> **R5**: We are deeply sorry that our submission may lead you to some misunderstandings. **In lines 199-201, our assertion is not that poisoned samples are universally 'easier' samples. Instead, we emphasize that these samples become 'easier' only when leveraging representations from the lower layers.** Given that existing text backdoor triggers inevitably leave abundant information at the word, phrase, or syntactic level, these signal is more pronounced at the lower layer [3]. In the deeper layers, the embeddings predominantly carry semantic features, which diminishes the backdoor signal, and the poison sample becomes 'harder' to learn [2].
>
> ---
>
> **Q6**: What type of model did you use to train the probing classifiers in Section 3? Is it a simple linear model?
>
> **R6**: Thank you for raising this question. In Section 3, we maintain consistency with our honeypot design for the probing classifiers. Specifically, we use a structure that consists of one transformer layer coupled with a simple linear classifier. While a simple linear model could potentially be sufficient, we plan to explore this in future work.
>
> ---
>
> **Reference**
>
> [1] Li, Yige, et al. "Anti-backdoor learning: Training clean models on poisoned data." NeurIPS 2021.
>
> [2] Biru Zhu, et al. "Moderate-fitting as a natural backdoor defender forpre-trained language models." NeurIPS 2022.
>
> [3] Ganesh Jawahar, et al. "What does bert learn about the structure of language? ACL 2019.
>
> [4] Zeng, Yi, et al. "Narcissus: A practical clean-label backdoor attack with limited information." arXiv 2022.
>
> [5] Xiangyu Qi, et al. "Revisiting the assumption of latent separability for backdoor defenses."" ICLR, 2023.

---

> > ### Comment · Reviewer_hYyY · 2023-08-17
> > **Thank you for your detailed response!**
> >
> > I'm satisfied with the ABL experiments and the poison ratio experiments, I'm increasing my score.
> >
> > There's a deeper truth here. ABL isolates easy samples by judging their training loss dynamics. Honeypot uses layer-wise dynamics. These two notions are correlated in the experiments for CV tasks (from Baldock et al.). Low training loss -> learned at an earlier layer easily. However, it seems like this is not the case for fine-tuned PLMs.
> >
> > You gave an intuitive answer to this phenomenon in your general comment but I think it justifies more experiments to understand why. To me, the most interesting result is in Supplement Figure 6. The poison samples become higher loss in deeper layers relative to clean samples. I would not expect that, and it's not surprising that ABL fails in this case (because it judges difficulty using the last layer).
> >
> > This brings me to a version of ABL that I think will work: attach a probe to an earlier layer and apply the loss-based sample filtering based on the loss from this probe (not from the model's last layer output).
> >
> > I would love to see a deeper, more systemic understanding of this fundamental separation from our prior understanding of CV models.

---

> > > ### Author Response · Authors · 2023-08-19
> > >
> > > We would like to sincerely thank you again for your time and valuable comments. We hereby provide more insights and explanations.
> > >
> > > ---
> > >
> > > **NQ1:** A new version of ABL that I think will work: attach a probe to an earlier layer and apply the loss-based sample filtering based on the loss from this probe (not from the model's last layer output).
> > >
> > > **NR1:** Thank you for the insightful suggestion! In fact, the "probe" method closely mirrors the core idea behind our honeypot :)
> > >
> > > Following your suggestion, we conducted initial experiments by placing the probe at the lower layers of PLMs for backdoor sample isolation. We denote this new variant as ABL++. As illustrated in Table 1, **using the lower layer probe has shown promising enhancements to ABL (with an ASR < 30%)**.
> > >
> > > These results suggest that **structural differences in backdoor signals can be integrated with existing defenses to further boost their effectiveness**, especially for pretrained models. We will explore this phenomenon more deeply in our revised manuscript.
> > >
> > > **Table 1: The comparison to ABL and ABL++**
> > > | Model$\downarrow$ | Dataset$\downarrow$ | Attack$\downarrow$, Defense$\rightarrow$  | ABL (ACC/ASR)  | ABL++ (ACC/ASR)  | Honeypot (ACC/ASR) |
> > > |:-----:|:-------:|:------:|:-----:|:---:|:---:|
> > > |RoBERTa$_{base}$|SST-2|AddWord| 90.25/76.21 |  93.00/22.57 | 93.71/6.65|
> > > |||AddSent| 91.17/69.24 | 92.01/15.07 | 92.39/7.71|
> > >
> > > ---
> > >
> > > **NQ2:** I would love to see a deeper, more systemic understanding of this fundamental separation from our prior understanding of CV models.
> > >
> > > **NR2:** Thank you for highlighting this interesting and important point!
> > >
> > > - Although this paper mainly focuses on pretrained language models in NLP, we also recognize the significance of CV tasks.
> > > - To further study this interesting problem, we conduct preliminary experiments on CIFAR-10 with ResNet-50 pretrained on ImageNet. The results show that the **pre-trained CV model also initially concentrates on learning task-related features before backdoor-related features**. In other words, **this interesting inconsistent behavior is due to different training paradigms** (i.e., fine-tune pre-trained model v.s. training from scratch) instead of different tasks. We will include this in our updated supplementary.
> > > - We argue that previous backdoor attack/defense work has not identified this phenomenon mostly because they usually train CV models (e.g., VGG and ResNet) on datasets (e.g., CIFAR-10) from scratch. Accordingly, we believe our observations are critical to future defense research since CV tasks are currently also embracing the training paradigm of fine-tuning large foundation models. We will further explore this problem in CV tasks in our future works.

---

### Official Review · Reviewer_5e6e · 2023-07-08

**Soundness:** 2 fair
**Presentation:** 2 fair
**Contribution:** 2 fair
**Rating:** 3
**Confidence:** 4

**Summary:**

This paper proposes a method to defend against NLP backdoors during training.
The proposed method works by using an honeypot module to absorb backdoor
information, and prevent the backdoor behaviors to be learned by the stem
network. Experiments on SST-2, IMDB, and OLID demonstrate the effectiveness of
the propsoed method.


**Strengths:**

* The investigated problem is interesting.

* The motivation of this paper is good.

**Weaknesses:**

* The proposed method is based on the observation that
learning the backdoor task is generally easier than learning
the main task. However, this observation may not always hold
true. In the case of label-specific backdoor attacks (also
called as all-to-all attak in BadNets [1]), where samples
with different original labels have different target labels,
the backdoor task becomes even more complex compared to the
main task. To achieve the desired backdoor behavior in such
attacks, the model must first identify the correct label of
the backdoor samples before making backdoor predictions
based on that recognized label. Unfortunately, this paper
lacks a discussion and empirical results concerning
label-specific attacks. As a result, the generalizability of
the proposed method to different types of attacks remains
unclear.

* Comparison to related work CUBE [2] is missing. While this
paper claims that the proposed method surpasses existing
defenses, it fails to include a comparison to CUBE, a
training-time textual backdoor defense method. It is
recommended to incorporate a comparison with CUBE to provide
a more comprehensive evaluation of the proposed method's
performance.


* The proposed approach essentially considers samples with
W(x) values below the threshold value c as identified
poisoning samples. To gain a deeper understanding of the
proposed method's effectiveness, it is recommended to
discuss the measures of detection precision and recall in
detail.

[1] Gu et al., BadNets: Identifying Vulnerabilities in the Machine Learning Model Supply Chain. arXiv 2017.

[2] Cui et al., A Unified Evaluation of Textual Backdoor Learning: Frameworks and Benchmarks. NeurIPS 2022 Datasets & Benchmarks.

**Questions:**

See Weaknesses.

**Limitations:**

The limitations is discussed in the Appendix.

---

> ### Author Rebuttal · Authors · 2023-08-10
>
> **Q1**: The proposed method is based on the observation that learning the backdoor task is generally easier than learning the main task. However, in the case of the all-to-all attack, where samples with different original labels have different target labels, the backdoor task becomes even more complex compared to the main task.
>
> **R1**: Thank you for this insightful comment!
>
> - In this paper, **we only consider all-to-one attacks simply following the settings used in almost all backdoor defenses in NLP**.
> - Arguably, **all-to-all attacks are less practical for attackers** since they are less controllable and harder to succeed compared to all-to-one attacks.
> - However, we do understand your concerns. Accordingly, we also compare defenses' performance under all-to-all attacks where the target label is set to $y' = (y + 1) \mod K$, where $K$ is the number of classes. We conduct experiments on SST-2 and AGNews datasets with the BERT model. As shown in the following table, **our method is better than chosen baseline defenses**. However, we also notice that **the performance of all defenses is significantly lower than that of the case in defending against all-to-one attacks**. This is because these attacks are harder to learn, as you suggest, and thus their attack signals are weaker for defenses to catch. We will explore how to better defend against them in our future work.
>
> **Table 1: Results on all-to-all attack**
> | Model | Dataset | BKI (ACC/ASR) | Onion (ACC/ASR) | CUBE (ACC/ASR) | Honeypot (ours) (ACC/ASR) |
> |:-----:|:-------:|:-----:|:-----:|:-----:|:-----:|
> |BERT$_{base}$|SST-2|91.62/91.05|89.25/69.24|90.12/59.09| **89.10/46.21** |
> ||AGNews|92.69/88.69|89.64/63.20|91.39/58.19|**90.67/39.78**|
> |BERT$_{large}$|SST-2|93.11/89.44|92.75/79.18|92.36/51.35|**91.85/46.67**|
> ||AGNews|93.40/91.22|91.58/70.32|93.01/60.10|**91.42/41.44**|
>
> ---
>
>
>
> **Q2**:  It is recommended to incorporate a comparison with CUBE [1] to provide a more comprehensive evaluation of the proposed method's performance.
>
> **R2**: Thank you for this constructive suggestion! To further alleviate your concerns, we compare our method with CUBE on SST-2 and IMDB datasets under AddWord and AddSent. The results (from Table 1 in our General Response) show that **our method is more effective (with lower ASR)**. We will provide more details and experiments in the appendix of our revision.
>
> ---
>
> **Q3:** The proposed approach essentially considers samples with W(x) values below the threshold value c as identified poisoning samples. To gain a deeper understanding of the proposed method's effectiveness, it is recommended to discuss the measures of detection precision and recall in detail.
>
> **R3**: Thank you for this constructive suggestion!
> - In general, different from the backdoor unlearning method [2], which requires to detect of a small portion of poison samples, **our honeypot module intends to capture almost all poisoned samples (i.e., have a high recall) instead of correctly identifying poison samples (i.e., high precision)**. It is because poisoned samples are highly effective, and even a few remaining ones can still create hidden backdoors during model training.
> - Following your suggestions, we hereby calculate the precision and recall of detecting poisoned samples generated by AddWord and StyleBKD on the SST2 dataset. As shown in the following Table 2, **our method can consistently achieve high recall while maintaining reasonable precision across diverse thresholds**. Although the precision might not be as consistently high, the main experiments in our paper indicate that reducing the learning weight for those clean samples with low loss has minor effects in reducing the performance of the original task.
>
>
>
> **Table 2: The precision and recall of our honeypot module in detecting poisoned samples.**
> | c$\downarrow$ | AddWord (Precision / Recall)      |  AddSent (Precision / Recall)    |
> |:---------:|:------------:|:-------------:|
> | 0.05 | 58.50 / 87.71 | 53.29 / 90.46 |
> | 0.1 | 32.60 / 99.28 | 34.37 / 99.54 |
> | 0.2 | 18.79 / 99.58 | 15.21 / 99.87 |
>
> **Reference**
>
> [1] Cui, Ganqu, et al. "A unified evaluation of textual backdoor learning: Frameworks and benchmarks." NeurIPS 2022.
>
> [2] Li, Yige, et al. "Anti-backdoor learning: Training clean models on poisoned data." NeurIPS 2021.

---

> > ### Author Response · Authors · 2023-08-19
> >
> > Dear Reviewer 5e6e,
> >
> > Thank you once again for your valuable time and constructive comments. We would like to kindly inform you that we have addressed your concerns in our rebuttal by: **(1)** evaluating with the all-to-all attack,  **(2)** comparing to more baselines, and **(3)** providing the results of detection precision/recall.
> >
> > As the reviewer-author discussion phase is nearing to the end, we want to check in with you to see if you have any further questions or concerns regarding our response. We are more than happy to answer any additional questions during the post-rebuttal period. Your feedback will be greatly appreciated.

---

> > ### Comment · Reviewer_5e6e · 2023-08-20
> >
> > Thanks for your responses. After reading the rebuttal, my concerns about the underlying assumption of this paper (learning the backdoor task is easier than learning the main task) still remains.
> >
> > 1. I don't think all-to-all attack is absolutely less practical. Compared to all-to-one attack, all-to-all attack can make the model predict different target labels based on the selection of the original labels, while one all-to-one trigger only has one fixed target label. Consider a sentiment classification scenario, the all-to-all trigger can flip the positive prediction to negative and also convert the negative prediction to positive, while the all-to-one trigger is only associate to one of the sentiment. In this case, all-to-all attack is actually more practical and powerful than the all-to-one trigger. Also, the attack success rates of the all-to-all attack on the well-trained models are typically above 90%, which are high enough. At least, all-to-all attack is an important type of attacks in the machine learning backdoor field.
> >
> > 2. There are many NLP backdoor defenses that are able to defend against all-to-all attack, such as Liu et al. and Shen et al. NIST TrojAI competition (https://pages.nist.gov/trojai/) also includes all-to-all attack and label-specific attack in the NLP rounds.
> >
> > 3. This paper claims the proposed method is robust to the adaptive attack. However, the all-to-all attack is actually a simple yet effective adaptive attack that diminish the performance of the proposed attack significantly, which makes the claim (i.e., robustness on the adaptive attack) might not be entirely accurate. The observations and the assumptions behind the proposed method is theoretically contradict to the ability for defending against all-to-all attack or label-specific attack, which might be a nonnegligible  technical flaw.
> >
> > Based on these concerns, I keep my score.
> >
> > Liu et al., "PICCOLO : Exposing Complex Backdoors in NLP Transformer Models" in IEEE Symposium on Security and Privacy 2022.
> >
> > Shen et al., "Constrained Optimization with Dynamic Bound-scaling for Effective NLPBackdoor Defense" in ICML 2022.

---

> > > ### Author Response · Authors · 2023-08-21
> > > **Rebuttal regarding all-to-all attack by Authors**
> > >
> > > Thank you for your insightful feedback. In light of them, we'd like to provide a clearer explanation to address potential misconceptions regarding our defense setting and further alleviate your concerns for the all-to-all attack.
> > >
> > > - We deeply appreciate your in-depth analysis of the all-to-all attack's practicality. Indeed, we recognize and value the inherent advantages of all-to-all attacks, especially in specific contexts like sentiment classification.
> > > - It appears that there might be some misunderstandings regarding our threat model. The referenced papers [1,2] are primarily tailored for backdoor detection tasks, where given a suspect model and a handful of benign samples, the aim is to detect or reverse-engineer the backdoor trigger. This inverted trigger is then utilized for backdoor unlearning. Yet, as outlined in our paper (lines 114-116), **our method diverges notably from these two-stage backdoor removal efforts. We don't rely on a clean dataset, instead advocating for a training-time defense where the model remains benign even when trained on a tainted dataset**. Most studies under our threat model, such as [3-7], primarily delve into the all-to-one attack, and our preliminary evaluations (as outlined in Table 1) suggest that existing defense approaches struggled to counter the all-to-all attack effectively.
> > > - To further address your valid concerns, we conducted experiments and revealed an interesting phenomenon: the all-to-all attack actually enhances our study's findings. Specifically, **we found a clear two-stage learning process that PLMs initially concentrate on task-related features and then shifting their attention to backdoor features**. This behavior stems from the inherent nature of the all-to-all attack: models need to first learn the primary task before delving into the backdoor task. This clear separation led us to suggest a straightforward solution: employing early stopping to prevent the model from learning the backdoor features. **As depicted in Table 1, implementing early stopping substantially reduce the ASR to around 10% while only marginally affecting original task performance**. These initial findings indicate that within the NLP realm, the all-to-all attack might not be as formidable as presumed. Furthermore, our method still shows the ability for defending against all-to-all attack. We're eager to delve deeper into all-to-all attacks and conduct more extensive experiments in the future.
> > >
> > > Once again, we're grateful for your valuable insights, and we hope this offers a clearer perspective on our work.
> > >
> > > **Table 1: Results on all-to-all attack**
> > > | Model | Dataset | BKI (ACC/ASR) | Onion (ACC/ASR) | CUBE (ACC/ASR) | Honeypot (ACC/ASR) | Honeypot+Early stopping (ACC/ASR) |
> > > |:-----:|:-------:|:-----:|:-----:|:-----:|:-----:|:-----:|
> > > |BERT$_{base}$|SST-2|91.62/91.05|89.25/69.24|90.12/59.09|89.10/46.21|88.64/14.33|
> > > ||AGNews|92.69/88.69|89.64/63.20|91.39/58.19|90.67/39.78|90.73/9.61|
> > > |BERT$_{large}$|SST-2|93.11/89.44|92.75/79.18|92.36/51.35|91.85/46.67|90.36/9.63|
> > > ||AGNews|93.40/91.22|91.58/70.32|93.01/60.10|91.42/41.44|90.69/7.25|
> > >
> > > **References**
> > >
> > > [1] Yingqi Liu, Guangyu Shen, et al., "PICCOLO : Exposing Complex Backdoors in NLP Transformer Models" in IEEE Symposium on Security and Privacy 2022.
> > >
> > > [2] Guangyu Shen, Yingqi Liu, et al., "Constrained Optimization with Dynamic Bound-scaling for Effective NLPBackdoor Defense" in ICML 2022.
> > >
> > > [3] Yige Li, Xixiang Lyu, et al. "Anti-backdoor learning: Training clean models on poisoned data." NeurIPS 2021
> > >
> > > [4] Fanchao Qi, Yangyi Chen, et al. "ONION: A Simple and Effective Defense Against Textual Backdoor Attacks." EMNLP 2021
> > >
> > > [5] Ganqu Cui, Lifan Yuan, et al. "A unified evaluation of textual backdoor learning: Frameworks and benchmarks." NeurIPS 2022.
> > >
> > > [6] Biru Zhu, Yujia Qin, et al. "Moderate-fitting as a natural backdoor defender forpre-trained language models." NeurIPS 2022
> > >
> > > [7] Xiangyu Qi, Tinghao Xie, et al. "Revisiting the assumption of latent separability for backdoor defenses." ICLR, 2023.

---

> > > > ### Comment · Reviewer_5e6e · 2023-08-21
> > > >
> > > > Thanks for your further responses. Below are my remaining concerns
> > > >
> > > > 1. Considering the backdoor defenses sharing the same threat model as this paper, there are also many methods having the capability to defend against all-to-all attacks, such as Chen et al. and Jin et al. I know some of them mainly conduct experiments on CV tasks, but they might be able to be adopted in the NLP tasks.
> > > >
> > > > 2. Although the results in Table 1 reflect that Honeypot outperforms existing methods. It is important to note that the design of these methods (such as CUBE) does not contradict the ability for defending against all-to-all attacks, which means it is possible to further improve their robustness on all-to-all attacks. My concern about the theoretical contradiction between the assumption for design and the robustness on all-to-all attack still remains.
> > > >
> > > > 3. The authors claim that they find a new phenomenon and proposed a new strategy that can be combined with the Honeypot and achieve satisfying results on all-to-all attacks. The proposed observations and the early-stopping strategy are highly similar to the findings and approaches introduced by Zhu et al. What are the fundamental differences between the proposed method and Zhu et al.? Also, the details about how Honeypot is combined with the early-stopping are missing. The new results in Table 1 showcase that the early-stopping strategy similar to Zhu et al. can effectively defend against all-to-all attack, but the robustness of Honeypot might be still unconvincing.
> > > > What are the results of using the early-stopping strategy solely?
> > > >
> > > > Chen et al., Effective Backdoor Defense by Exploiting Sensitivity of Poisoned Samples. NeurIPS 2022.
> > > >
> > > > Jin et al., Incompatibility Clustering as a Defense Against Backdoor Poisoning Attacks. ICLR 2023.
> > > >
> > > > Zhu et al. Moderate-fitting as a Natural Backdoor Defender for Pre-trained Language Models. NeurIPS 2022.

---

> > > > > ### Author Response · Authors · 2023-08-21
> > > > >
> > > > > Thank you again for your further comments, and we do understand your concerns. We hereby provide more details and explanations to further alleviate your remaining concerns, as follows.
> > > > >
> > > > > - We appreciate the reviewer for highlighting two more defense approaches [1,2]. **While both papers are indeed interesting and seem like they could be directly used in our task, their applications to the NLP domain are actually challenging.** For example, Chen et al. [1] heavily depended on image transformations (e.g., rotation and scaling) for detecting poisoned samples, which cannot be directly exploited in NLP tasks. We will further explore how to properly generalize these methods to NLP tasks in future work.
> > > > >
> > > > > - There might be a potential misunderstanding: although all-to-all attacks are more complex than all-to-one attacks, **it doesn't imply that the complexity of the all-to-all backdoor task surpasses the main task in shallow layers.** Our prior experiments demonstrated that the proposed method achieved competitive performance against baselines (with an ASR lower than 50%), proving that there's no theoretical contradiction between the assumption for our method and the robustness on all-to-all attack. Furthermore, we've discovered that the **incorporation of a simple early stopping mechanism is highly effective in mitigating all-to-all attacks, leading us to believe that these attacks are not a significant obstacle to our method.** Assessing the inherent difficulty of the main task and backdoor task is an interesting direction, and we're keen to explore this in future research.
> > > > >
> > > > >
> > > > >
> > > > > - We are deeply sorry that our previous explanations may lead you to some misunderstandings.
> > > > >     - **We did not argue that early stopping strategy is a novel method and should be regarded as our main contribution.** Instead, we intended to demonstrate **a simple strategy that can significantly bolster the performance of our method against all-to-all attacks in NLP tasks**. In other words, all-to-all attacks will not be a hindrance to our honeypot defense. Notice that this simple approach has been employed in many backdoor defense works [3-5].
> > > > >     - **The observation** that "PLMs initially focus on task-related features and later shift their attention to backdoor features" **wasn't newly presented in the rebuttal**. We have emphasized this finding in our manuscript, and a similar phenomenon was reported in [3].
> > > > >
> > > > > **References**
> > > > >
> > > > > [1] Weixin Chen, Baoyuan Wu, et al., Effective Backdoor Defense by Exploiting Sensitivity of Poisoned Samples. NeurIPS 2022.
> > > > >
> > > > > [2] Charles Jin, Melinda Sun, et al., Incompatibility Clustering as a Defense Against Backdoor Poisoning Attacks. ICLR 2023.
> > > > >
> > > > > [3] Biru Zhu, Yujia Qin, et al. Moderate-fitting as a Natural Backdoor Defender for Pre-trained Language Models. NeurIPS 2022.
> > > > >
> > > > > [4] Hongbin Liu, Jinyuan Jia, et al. PoisonedEncoder: Poisoning the Unlabeled Pre-training Data in Contrastive Learning. USENIX Security 2022.
> > > > >
> > > > > [5] Eric Wallace, Tony Zhao, et al. Concealed data poisoning attacks on nlp models. NAACL 2021.

---

### Author Rebuttal · Authors · 2023-08-10

# General Response to All Reviewers

We sincerely thank the reviewers for dedicating their time and providing invaluable feedback. We present a general reply below in response to the concerns raised regarding baseline comparisons.

**Q1**: Comparison with more baselines.

**R1**: Thank reviewers for this constructive suggestion!

- To address the reviewer's concerns, we compare our method with two additional baselines (i.e., CUBE [1] and ABL [2]). We conduct experiments on SST2 and IMDB datasets with the RoBERTa model, using AddWord and AddSent attacks. As shown in the following table, **our Honeypot method is more effective than baselines with lower ASR and higher ACC**.

**Table 1: The comparison to CUBE and ABL**
| Model$\downarrow$ | Dataset$\downarrow$ | Attack$\downarrow$, Defense$\rightarrow$  |  CUBE (ACC/ASR)  | ABL (ACC/ASR)  | Honeypot (ours) (ACC/ASR) |
|:-----:|:-------:|:------:|:-----:|:---:|:---:|
|RoBERTa$_{base}$|SST-2|AddWord| 92.32/17.34 | 90.25/76.21 | **93.71/6.65** |
|||AddSent| 92.48/27.25 | 91.17/69.24 | **92.39/7.71** |
||IMDB|AddWord| 91.58/28.76 | 92.59/87.14 | **93.72/5.60** |
|||AddSent| 92.12/36.41 | 89.75/88.77| **92.72/6.56** |
|RoBERTa$_{large}$|SST-2|AddWord| 94.09/18.67 | 92.03/74.98| **94.15/5.84** |
|||AddSent| 94.32/24.92 |91.77/67.05 | **94.83/4.20** |
||IMDB|AddWord| 93.68/17.85 | 92.59/75.09| **94.12/3.60** |
|||AddSent| 93.50/23.88 | 89.07/90.54| **93.68/6.32** |

- **Explaining the Results of ABL:** In Table 1, we found that ABL only achieves disappointing results with an ASR higher than 70%. To shed light on this outcome of ABL, we assessed the backdoor isolation capabilities of ABL. Following the setting in the ABL paper, we initiated a hyperparameter search where $\gamma$ denotes the loss threshold and $T_{te}$ stands for the epochs of the backdoor isolation stage. Table 2 presents the detection precision of the 1% isolated backdoor examples, which is crucial for the ABL backdoor unlearning performance. However, **our findings reveal that the percentage of poisoned samples is less than 20%, which accounts for ABL's suboptimal performance.**

**Table 2: The isolation precision (%) of ABL.**
| $\gamma$ $\downarrow$ $T_{te}$ $\rightarrow$ | 1 epoch | 5 epochs | 10 epochs |
|:---:|:---:|:---:|:---:|
|0.5| 2.1 | 11.7 | 13.5 |
|1.0| 5.1| 12.3 | 15.3 |
|1.5| 5.5 | 12.4 | 15.6|

---

**Q2:** Why does Honeypot perform better than ABL?

**R2:** The ABL method primarily relies on the observation that 'models learn backdoored data much faster than they do with clean data' [2]. However, it is crucial to note that **this assumption mainly holds for models trained from scratch in computer vision tasks**. Our research and reference [3] both demonstrated an opposite behavior that **pre-trained language models first concentrate on learning task-specific features before backdoor features**. A plausible explanation for this behavior is the richness of semantic information already present **in the top layers** of the pre-trained language models. Thus, the original task becomes more straightforward compared to the backdoor functionality, causing the model to prioritize learning the main task first. As a result, ABL struggles to yield satisfactory detection performance during the backdoor isolation stage by selecting the "easy-to-learn" samples (as shown in Table 2), and we show that ABL obtains a high ASR in the following backdoor unlearning process (as shown in Table 1). In contrast, our findings underscore the significance of examining model structure when identifying backdoor samples, revealing that backdoor samples become more identifiable in the lower layers of PLMs.

**Reference**

[1] Cui, Ganqu, et al. "A unified evaluation of textual backdoor learning: Frameworks and benchmarks." NeurIPS 2022.

[2] Li, Yige, et al. "Anti-backdoor learning: Training clean models on poisoned data." NeurIPS 2021.

[3] Biru Zhu, et al. "Moderate-fitting as a natural backdoor defender forpre-trained language models." NeurIPS 2022.

---

### Decision · Program_Chairs · 2023-09-21

**Decision:**

Accept (poster)

**Comment:**

This paper proposes a new training-time defense method against backdoor for pretrained language models. The idea is that suspicious poisoning data tend to have more discriminative feature representation in early layers. By training a classifier on early layer features, one can dynamically identify suspicious/trustworthy data and reweight them during training. This effectively mitigates the attack during training.

Reviewers generally agree that the idea is novel and effective. The experiments are comprehensive and convincing. Reviewers asked for experiments on previous baselines which also dynamically adjust sample weights. In rebuttal, the authors provided these baselines and discussed the results carefully.

One reviewer raised concern regarding the setting, i.e., the paper did not prove its defensive power against all-to-all attacks. While we do agree that all-to-all is an important attack scenario, considering the large flexibility of attacks, it seems OK to overlook this restriction and focus on the innovation and efficacy of the proposed method.